# Systematic imaging reveals features and changing localization of mRNAs in *Drosophila* development

Helena Jambor[1]*, Vineeth Surendranath[1], Alex T Kalinka[1,2], Pavel Mejstrik[1], Stephan Saalfeld[1,3], Pavel Tomancak[1]*

[1]Max Planck Institute of Molecular Cell Biology and Genetics, Dresden, Germany; [2]Institute of Population Genetics Department of Biomedical Sciences, University of Veterinary Medicine Vienna, Vienna, Austria; [3]Janelia Research Campus, Howard Hughes Medical Institute, Ashburn, United States

**Abstract** mRNA localization is critical for eukaryotic cells and affects numerous transcripts, yet how cells regulate distribution of many mRNAs to their subcellular destinations is still unknown. We combined transcriptomics and systematic imaging to determine the tissue-specific expression and subcellular distribution of 5862 mRNAs during *Drosophila* oogenesis. mRNA localization is widespread in the ovary and detectable in all of its cell types—the somatic epithelial, the nurse cells, and the oocyte. Genes defined by a common RNA localization share distinct gene features and differ in expression level, 3′UTR length and sequence conservation from unlocalized mRNAs. Comparison of mRNA localizations in different contexts revealed that localization of individual mRNAs changes over time in the oocyte and between ovarian and embryonic cell types. This genome scale image-based resource (Dresden Ovary Table, DOT, http://tomancak-srv1.mpi-cbg.de/DOT/main.html) enables the transition from mechanistic dissection of singular mRNA localization events towards global understanding of how mRNAs transcribed in the nucleus distribute in cells.

*For correspondence: jambor@mpi-cbg.de (HJ); tomancak@mpi-cbg.de (PT)

**Competing interests:** The authors declare that no competing interests exist.

**Reviewing editor**: Karsten Weis, University of California, Berkeley, United States

## Introduction

Cell differentiation is accompanied by polarization and segregation of membranes, cytoplasm, and organelles. A powerful mechanism to generate subcellular asymmetries used by eukaryotes and even prokaryotes is mRNA localization in combination with controlled protein translation (reviewed in *Medioni et al., 2012*). Long-range mRNA transport in most metazoans relies on the polarized cytoskeleton and the microtubule minus- and plus-end motor complexes. mRNA enrichment at microtubule minus-ends is aberrant in mutants that affect the dynein motor complex, while plus-end directed transport requires kinesin molecules (reviewed in *Bullock, 2011*; *Medioni et al., 2012*)

Mechanistic dissection of several canonical localization examples showed that, mRNAs localize through *cis*-regulatory sequences, zipcodes, which are often present in the 3′UTR of the transcript (reviewed in *Jambhekar and Derisi, 2007*) and zipcode-binding proteins that initiate the formation of transport competent ribonucleoproteins (RNPs) (*Dienstbier et al., 2009*; *Bullock et al., 2010*; *Chao et al., 2010*; *Dix et al., 2013*). mRNAs can also harbour two antagonizing localization signals that act consecutively in cells and direct mRNAs sequentially to opposing microtubule ends (*Ghosh et al., 2012*; *Jambor et al., 2014*), suggesting that transport RNPs could be regulated. It has further been shown that some mRNA localization elements are active in several cell types suggesting that the mRNA transport machinery is widely expressed and mRNA localization elements function in a cell-type independent manner (*Kislauskis et al., 1994*; *Bullock and Ish-Horowicz, 2001*; *Snee et al., 2005*; *Jambor et al., 2014*).

**eLife digest** To make a protein, the DNA sequence that encodes it must first be 'transcribed' to build a molecule of messenger RNA (called mRNA for short). Although many mRNA molecules are found throughout a cell, some are 'localized' to certain areas; and recent evidence suggests that this mRNA localization may be more common than previously thought.

Not much is known about how cells identify which mRNAs need to be localized, or how these molecules are then transported to their destination. The localization process has been studied in most detail in the developing egg cell—also known as an oocyte—of the fruit fly species *Drosophila melanogaster*. These studies have identified few mRNA molecules that, if they are not carefully localized within the cell, cause the different parts of the fly embryo to fail to develop correctly when the oocyte is fertilized.

Jambor et al. created an open-access online resource called the 'Dresden Ovary Table' that shows how 5862 mRNA molecules are distributed in several cell types involved in oocyte production in the ovary of female *D. melanogaster* flies. This resource consists of a combination of three-dimensional fluorescent images and measurements of mRNA amounts recorded at different stages in the development of the oocyte.

Using the resource, Jambor et al. demonstrate that all of the cell types that make up the ovary localize many different mRNA molecules to several distinct destinations within the cells. The localized mRNAs share certain features, with mRNAs localized in the same part of the cell showing the most similarities. For example, localized mRNAs have longer so-called 3′ untranslated regions (3′UTR) that carry regulatory information and these sequences are also more evolutionarily conserved. Further, when the mRNA molecules in the oocyte were examined at different times during its development and compared with the embryo, the majority of these mRNAs were found to change where they are localized as the organism develops.

The resource can be used to gain insight into specific genetic features that control the distribution of mRNAs. This information will be instrumental for cracking the 'RNA localization code' and understanding how it affects the activity of proteins in cells.

In addition to microtubule-based transport, some mRNAs can enrich by trapping to a localized anchoring activity (*Forrest and Gavis, 2003*; *Sinsimer et al., 2011*) or by hitch-hiking along with a localization-competent mRNA (*Jambor et al., 2011*). Recent live-imaging studies revealed that the same mRNA can, depending on the cell type, use both diffusion and active transport mechanisms (*Park et al., 2014*). Furthermore, in vitro data showed that mRNA transport along microtubules can occur both uni- and bi-directionally, suggesting mRNAs can switch between processive and diffusive transport modes (*Soundararajan and Bullock, 2014*).

mRNA localization is perhaps best characterized in the oocyte of *Drosophila melanogaster* (*D. melanogaster*) where localization of *oskar*, *bicoid*, and *gurken* is instrumental for setting up the embryonic axes (*Berleth et al., 1988*; *St Johnston et al., 1989*; *Ephrussi et al., 1991*; *Neuman-Silberberg and Schüpbach, 1993*). However, more recent work suggests that mRNA localization is not occurring only for few singular mRNAs but instead is a widespread cellular feature that affects a large proportion of expressed mRNAs (*Shepard et al., 2003*; *Blower et al., 2007*; *Lecuyer et al., 2007*; *Zivraj et al., 2010*; *Cajigas et al., 2012*). How a cell distinguishes localized from ubiquitous transcripts and orchestrates transport of many mRNAs remains enigmatic. It is conceivable that each localized mRNA carries its own zipcode sequence that directs it to a specific subcellular location. However, despite wealth of data on co-localized transcripts, computational methods thus far fail to detect such signals in a reliable manner. Alternatively co-packaging of several mRNA species, only one of which carries specific localization signal, has been shown in at least two cases (*Lange et al., 2008*; *Jambor et al., 2011*). It is also unclear to what extent the mRNA localization status is subject to tissue specific regulation.

Here, we describe a genome-wide image-based resource that unravels the global landscape of mRNA localization in the *Drosophila* ovary by combining stage-specific mRNA sequencing with systematic fluorescent in situ hybridizations (FISH) and imaging. The localized transcripts show characteristic gene level features, such as longer and highly conserved 3′UTRs, which clearly distinguish subcellular enriched from ubiquitous mRNAs. Comparing mRNA localizations across the sampled time-points showed that the

localization status of the majority of mRNAs changes in the oocyte as oogenesis progresses. These changing localizations are not due to alternative gene expression since the germline cells of the *Drosophila* ovary show only little transcriptional change. Integrative analysis of ovary localization data together with similar data from embryos (*Lecuyer et al., 2007*) also revealed that mRNA localizations differ across cell types. Therefore, mRNA localization is widespread in cells and is highly regulated.

## Results

### Widespread mRNA localization in *Drosophila* ovaries

To globally investigate post-transcriptional regulation through mRNA localization, we systematically probed and imaged the expression and subcellular distributions of mRNAs in egg-chambers mass isolated from *Drosophila* ovaries. We combined stage-specific mRNA sequencing (3Pseq and RNAseq) with genome-wide fluorescent in situ hybridization (FISH). RNA sequencing data, expression pattern annotations (using a hierarchical controlled vocabulary-http://tomancak-srv1.mpi-cbg.de/cgi-bin-public/ovary_annotation_hierarchy.pl) and images (representative 2D images and all original z-stacks) are collected in a publicly accessible database, the Dresden Ovary Table, DOT (http://tomancak-srv1.mpi-cbg.de/DOT/main) (*Figure 1—figure supplement 1A,B*). This genome-wide resource also integrates data on tissue-specific gene expression (*Tomancak et al., 2002, 2007*) and subcellular mRNA localization (*Lecuyer et al., 2007*) in *Drosophila* embryos.

Based on our in situ hybridization screen, we identified 3475 mRNAs as being expressed and most of these mRNAs were also detectable by RNA sequencing. Both sequencing techniques were in good agreement with each other (*Figure 1A*, *Figure 2—figure supplement 1A*, *Figure 5—figure supplement 1A*). Of the expressed genes, 64% showed ubiquitous mRNA distribution in ovary cells throughout oogenesis (*ubiquitous*), but we also observed mRNA expressions restricted to subsets of cells (*cellular*) and mRNAs that asymmetrically localized in the cytoplasm (*subcellular*) or to the nuclei of cells (*nuclear*).

Subcellular mRNA localization affected 790 mRNAs (22%) but was limited to small number of subcellular domains (*Figure 1B–C*). The largest group was 591 mRNAs that were enriched in the oocyte portion of the syncytial egg-chamber during early oogenesis (*fwe, Imp, Shroom*). At this stage, the microtubule minus ends of the polarized microtubule cytoskeleton are also concentrated in the oocyte (reviewed in *Steinhauer and Kalderon, 2006*). At mid-oogenesis the oocyte establishes its own polarized microtubule cytoskeleton (*Steinhauer and Kalderon, 2006*) and at this stage, we observed 106 mRNAs enriched towards the anterior and 119 mRNAs enriched at the posterior pole. The quality of these localizations ranged from tight (*mus210, Lcp65Ac*) to diffuse association (*yemalpha, fs(1)N*) at the anterior-dorsal, the entire anterior or the posterior cortex. mRNAs were also detected in subcellular domains of the nurse (*msk, spoon*) and somatic epithelial cells (*CG43693, CG12171*). For few mRNAs, we observed previously unknown ovary accumulations, for example mRNAs in cytoplasmic granules (*CG17494*), depleted from the oocyte (*Nacalpha*), showing cortical enrichment (*Actn*), or forming ring-like structures (*CG14639*, *Figure 1B'*, *Figure 2—figure supplement 1E*).

The 309 mRNAs (13%) of the cellular category were predominantly expressed in the somatic epithelium (follicle cells) and often restricted to a subset of epithelial cells at specific oogenesis stages (*Figure 2A,B*). 191 RNAs were detectable specifically in ovarian nuclei, mostly of the endocycling, polyploid nurse cells, but also in epithelial cells and in 29 cases in the oocyte nucleus (*Figure 2C,D*). The RNAs in ovarian nuclei were visible from stage 9 of oogenesis onwards and their localization changed appearance from stage 9 to 10 (*Figure 2—figure supplement 1B*). Nuclear patterns varied from ring-like signal to dispersed foci or widespread distribution in the nucleoplasm and were not linked to the chromosomal position of the genes (*Figure 2—figure supplement 1C*). Precursors of micro RNAs and long non-coding RNAs also showed varying degrees of nuclear enrichments (*Figure 2—figure supplement 1D*).

In summary, our screen revealed countless new instances of tissue-specific gene expression and mRNA localization in the ovary. The relatively low number of different subcellular localization sites allowed us to group mRNAs into subcellular localization gene-sets containing tens to hundreds of co-regulated genes.

### Global features of localized mRNAs

The division of RNAs into gene sets enabled us to address whether genes within each class are functionally related (*Figure 1D*, *Figure 1—figure supplement 2*). Gene Ontology (GO) analysis

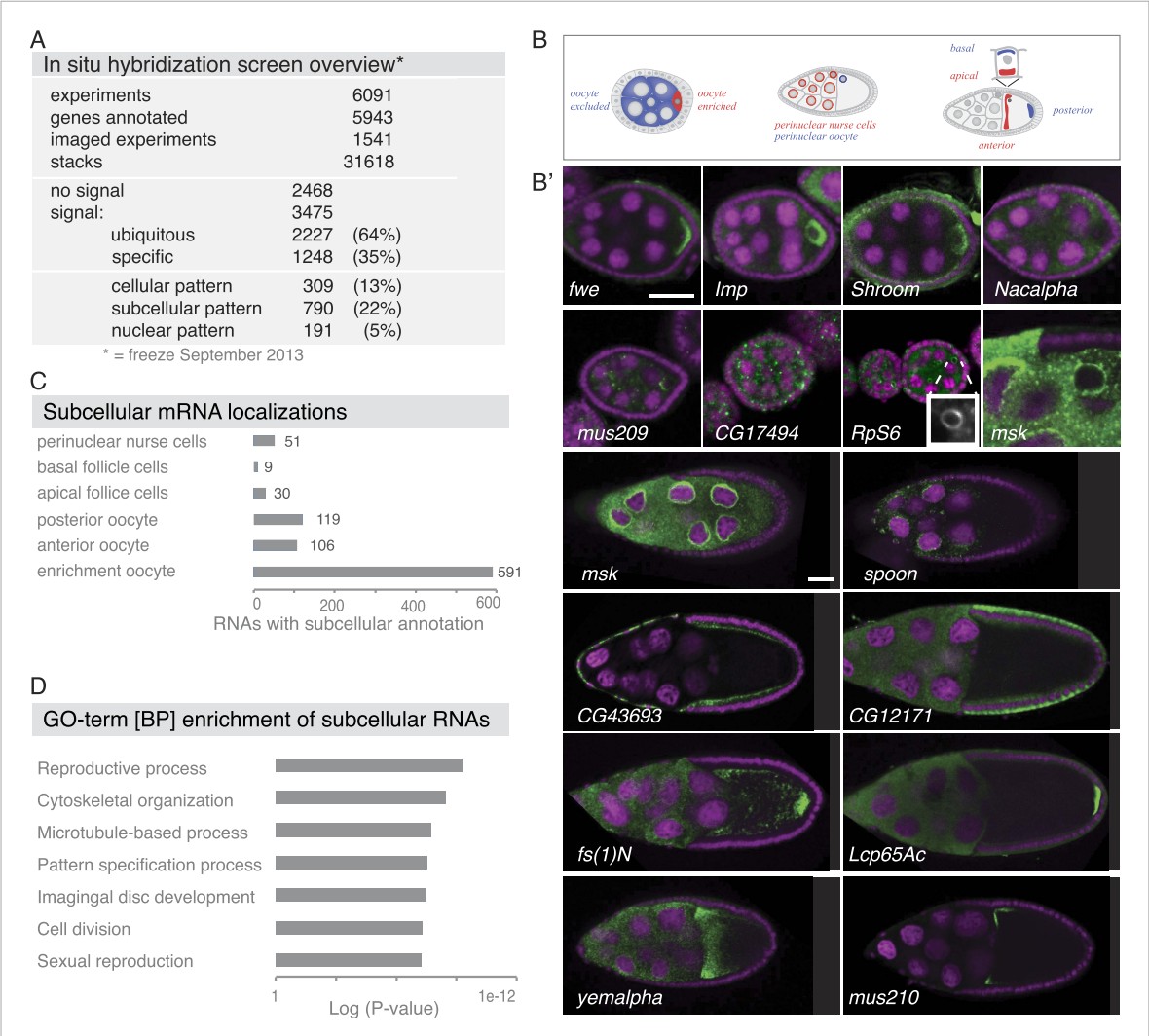

**Figure 1**. Summary of the fluorescent in situ hybridization (FISH) screen in ovaries. (**A**) Summary of key numbers of the screen. For each of the 6091 FISH experiments, we annotated the signal as no signal, ubiquitous, or specific. Specific and some ubiquitous signals were imaged. (**B**) Schematic of exemplary subcellular expression patterns. (**B'**) Exemplary subcellular expression patterns. In the syncytial early egg-chamber, 591 mRNAs are transported from the site of transcription in the nurse cells into the developing oocyte: mRNAs are either restricted to a cortical domain (*fwe*) or detectable in the entire ooplasm (*Imp*). mRNAs also simultaneously enriched in the oocyte portion of the syncytial egg-chamber and at the apical membrane of the somatic epithelial cells (*Shroom*). Five mRNAs were specifically excluded from the oocyte portion and enriched in the nurse cells (*Nacalpha*). Few mRNAs were enriched anterior in stage 2–7 oocytes (*mus209*). mRNAs showed ubiquitous granules in the cytoplasm (*CG17494*) or rarely ring-like staining patterns (*RpS6*, inset [10 × 10 µm] showing only the RNA channel). mRNAs enriched around the nucleus of the oocyte and/or the nurse cells (*msk*) varying from a ring around the entire nucleus to restricted localization in sub-areas of the perinuclear space (*spoon*). Apical enrichment (*CG43693*) or basal localization (*CG12171*) was detected in late epithelial somatic cells. Anterior and posterior RNA localization varied between diffuse (*fs(1)N, yemalpha*) and tight cortical enrichments (*Lcp65Ac, mus210*). (**C**) Distribution of subcellular localized mRNAs in subcategories. Note: mRNAs can appear in more than one subgroup. (**D**) GO-term enrichment analysis of ubiquitous, cellular, nuclear, and subcellular gene sets.

The following figure supplements are available for figure 1:

**Figure supplement 1**. Experimental outline and database features.

**Figure supplement 2**. GO-term enrichment analysis for gene sets.

showed that the subcellular gene set is distinct from the cellular and the nuclear gene sets. Consistent with their respective expression, cellular genes are enriched for epithelial development, lipid trafficking and cuticle formation, nuclear genes for RNA regulatory processes and the subcellular gene

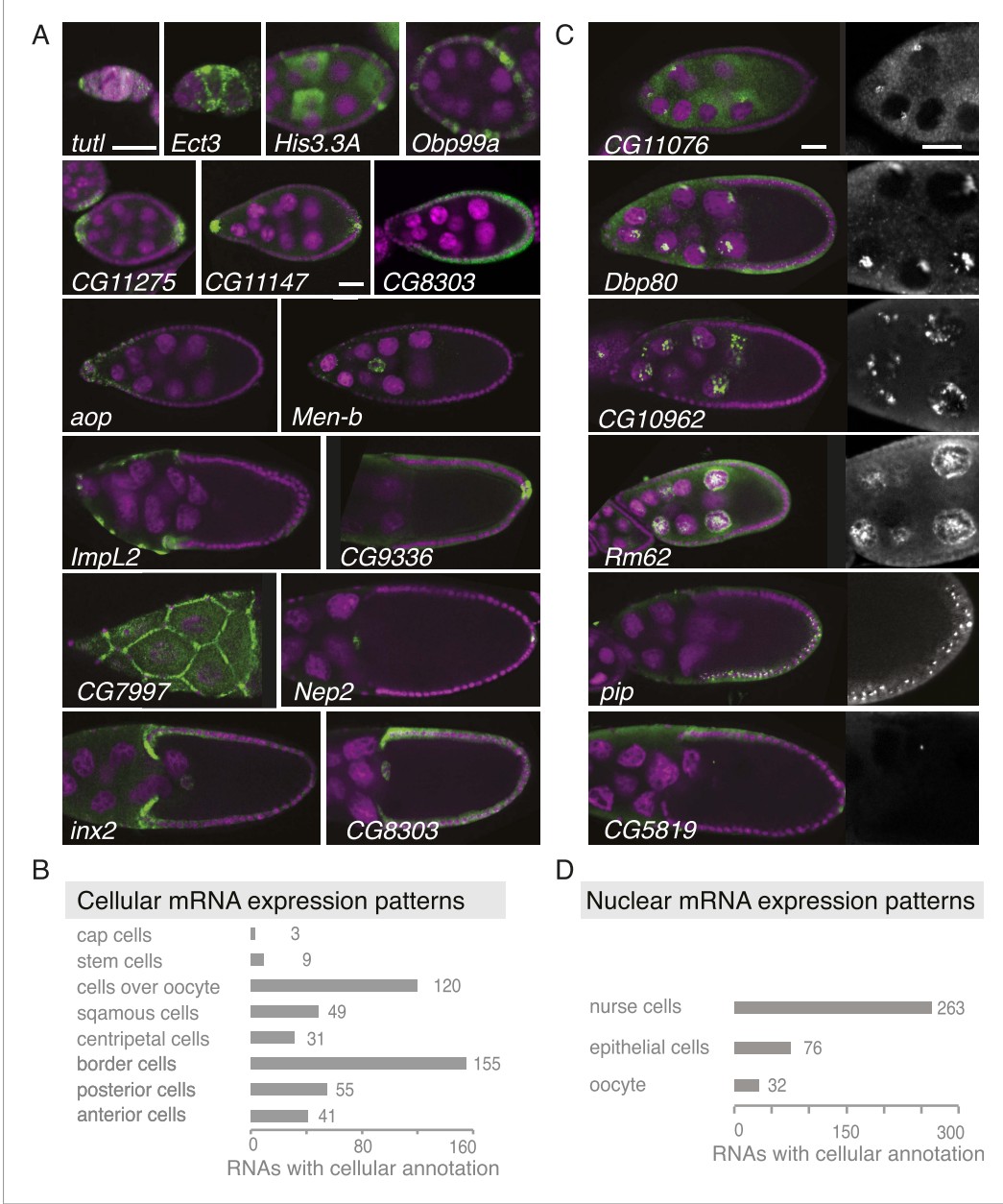

**Figure 2**. Summary of cellular and nuclear expression patterns. (**A,C**) Exemplary FISH experiments for the cellular (**A**) and nuclear (**C**) expression sets. RNA is shown in green and the DNA (labelled with DAPI) is shown in magenta. Scale bars: 30 µm. (**A**) *tutl* is expressed in cap cells at the tip of the germarium, while *Ect3* mRNA is detectable in the somatic epithelial cells of the germarium. Several mRNAs are expressed in mosaic pattern, indicating cell cycle control in somatic epithelial cells (*His3.3A, Obp99a*) and in nurse cells (*His3.3A*). Expression in the anterior and posterior follicle cells is often seen simultaneously (*CG11275, CG11147, Nep2*). Some mRNAs were expressed only in anterior follicle cells that become migratory border cells (*Men-b*) or in posterior follicle cells (*CG9336*). *CG8303* is expressed in the somatic cells destined to become columnar epithelium. *aop* is exclusively seen in follicle cells that will give rise to the squamous epithelium and several mRNAs are specifically expressed here at later stages (*ImpL2, CG7997*). mRNAs are also expressed in cells forming the border of columnar and squamous epithelial cells (*inx2*). (**C**) Nuclei enrichments of RNAs in nurse cells varies from a ring-like expression (*CG11076*) to foci in a discrete area (*Dbp80*), widespread foci (*CG10962*), or nucleoplasm signal (*Rm62*). RNAs are also detectable in epithelial cell nuclei (*pip*) and for 28 RNAs also in the oocyte nucleus (e.g., *CG5819*). Greyscale image shows the respective RNA staining only in a zoomed-in view. (**B**) The cellular gene set was subcategorized according to the specific cellular expression

*Figure 2. continued on next page*

*Figure 2. Continued*

pattern. Individual mRNAs can fall into several of these subgroups. (**D**) Instances of nuclear RNA enrichments in nurse cells, epithelial cells, and the oocyte.

The following figure supplement is available for figure 2:

**Figure supplement 1**. FISH screen results and controls.

---

set for reproductive processes, cytoskeleton organization, and cell cycle regulation. Anterior and posterior gene sets differed: anterior genes were enriched for microtubule terms and, being localized in proximity to the meiotic oocyte nucleus, are additionally associated with chromosome and cell cycle regulation terms. The posterior mRNAs associated strongly with signalling, cell fate commitment, and membrane organization terms. The GO analysis suggests that mRNAs that co-localize in the cytoplasm are functionally related.

We next asked whether the proteins encoded by the mRNAs show physical interactions. To this end, we analysed the protein interaction data (mentha interactome database [*Calderone et al., 2013*]), which revealed that proteins of the posterior gene set participate in significantly more protein–protein interactions than of the anterior gene set (*Figure 3B*). This suggests that the close proximity of their transcripts in the cell could be of functional importance. The gene sets defined by our ovary screen also maintained distinct expression patterns during embryogenesis. Genes of the subcellular sets are enriched among genes expressed in the central nervous system and epithelia, suggesting an interesting relatedness of these polarized tissues (*Figure 3A*, *Figure 3—figure supplement 1*). Thus, gene sets defined by ovary expression are co-regulated also beyond oogenesis.

We next investigated whether there are further global features that could set localized mRNAs apart from ubiquitous ones. Ovarian expressed mRNAs differed in their expression levels over several orders of magnitude. Using our stage specific 3Pseq data, we analysed the expression levels for each gene set. Ubiquitous and subcellular mRNA expression levels were overall comparable however, the posterior class was expressed significantly higher than all other localization classes, including the related anterior mRNAs (*Figure 3C–C'*, *Figure 3—figure supplement 2A*). Considering how seemingly inefficient posterior transport is (*Zimyanin et al., 2008*), higher expression levels could be an additional measure to ensure that enough mRNAs will eventually localize. In particular, the late phase accumulation of posterior localized mRNAs in the enlarged oocyte (*Forrest and Gavis, 2003*; *Sinsimer et al., 2011*) could benefit from high expression levels.

Yet, expression level alone cannot account for subcellular localization. We therefore compared the gene-level variables of each localization class and revealed that subcellular mRNAs had significantly longer 3′UTR sequences and this was more pronounced for the posterior localization class (*Figure 3D, D'*). The posterior gene set further showed longer gene structures, longer 5′UTRs, longer exons and introns, a higher number of exons and introns, and a higher intron proportion compared to ubiquitous and anterior mRNAs (*Figure 3—figure supplement 2B–H*). Consistent with the observation that localized mRNAs are enriched in non-coding portions, the exon proportion was the highest in the ubiquitous gene set (*Figure 3—figure supplement 2I*). The high intron proportion of posterior genes is particularly interesting in light of the recent finding that the stable deposition of the exon junction complex, required for posterior *oskar* mRNA localization, is correlated with long intron-containing genes (*Ashton-Beaucage et al., 2010*; *Ghosh et al., 2012*). Localized genes not only had longer 3′ UTRs, but also showed higher 3′UTR sequence conservation than ubiquitous genes, and again this was significantly more pronounced in the posterior gene set (*Figure 3E,E'*). We also observed longer and more conserved 3′UTRs in the embryo localized mRNAs (apical, posterior) compared to the embryo ubiquitous mRNAs (*Figure 3—figure supplement 3A,B* based on data from [*Lecuyer et al., 2007*]), indicating that these features are not specific to oocyte-localized mRNAs.

The posterior gene set shows clearly distinct functional and gene architectural features compared to all the other categories. We therefore decided to investigate whether the cytoplasmic localization of the novel candidate mRNAs depends on the known components of RNA localization machinery in the oocyte. First, we probed the dependency of mRNA localization on the microtubule cytoskeleton. Transport of known mRNAs towards the anterior and the posterior pole of the oocyte requires an intact microtubule cytoskeleton (reviewed in *Steinhauer and Kalderon, 2006*). We observed that

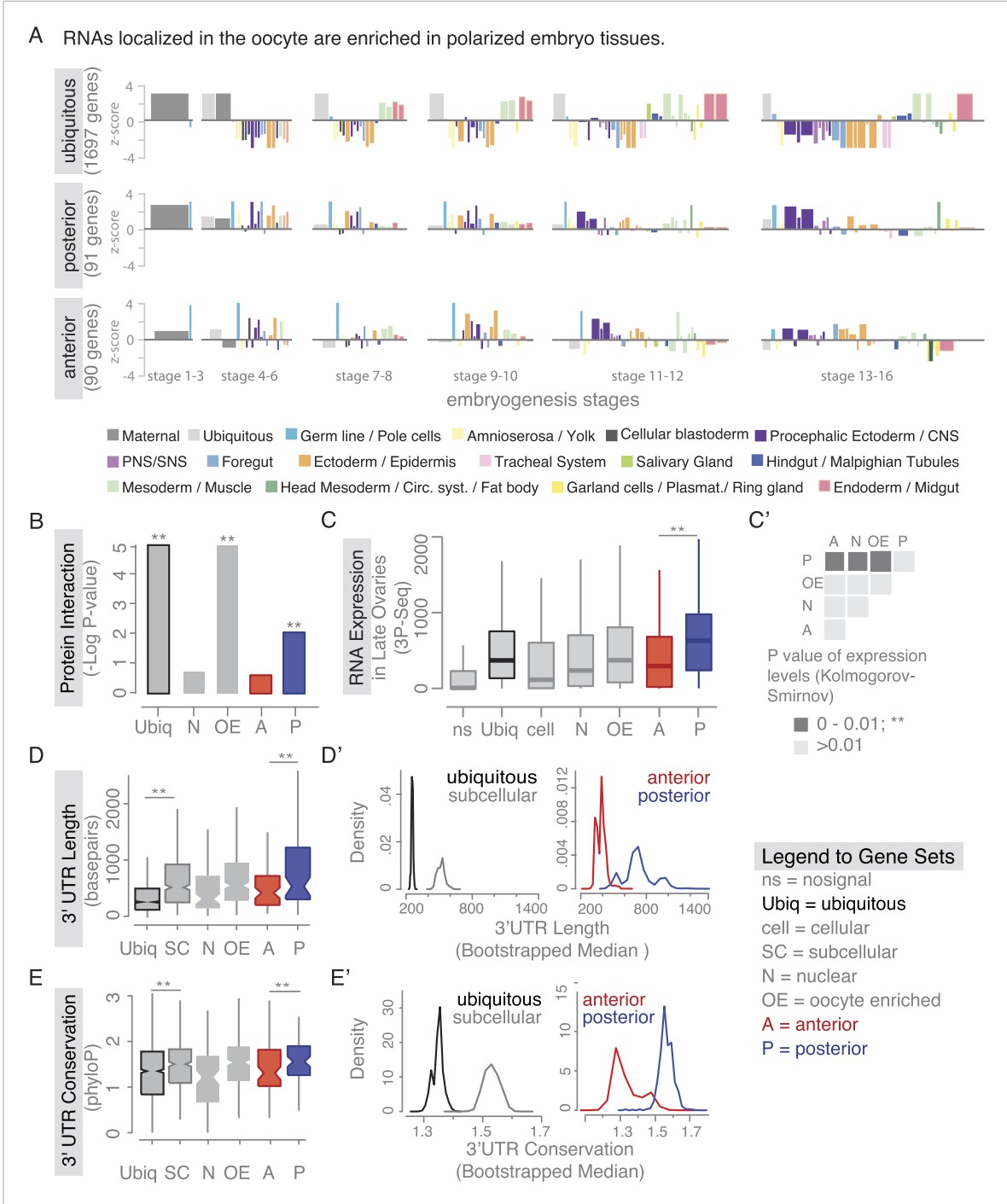

**Figure 3**. Localized mRNAs show gene set specific features. (**A**) Linear hierarchy plot (*Tomancak et al., 2007*) showing stage- and tissue-specific re-expression of the ovary gene sets in embryogenesis. (**B**) Protein interaction analysis per gene set revealed that posterior genes, but not anterior genes, share significantly more protein–protein interactions than would be expected by chance. (**C**) Boxplot showing the median mRNA expression level is significantly higher in the posterior gene set compared to anterior mRNAs (**C'**: Kolmogorov–Smirnov p-value: 3.9e-06). Shown are 3Pseq quantifications from late ovary mRNA (for early, full ovaries and early embryogenesis see *Figure 3—figure supplement 2A*). For description of gene sets see *Supplementary file 1*. (**D–E**) Distributions of median 3'UTR length (**D**) and conservation of the 3'UTR sequence (**E**, across 24 *Drosophila* species) for gene sets. (**D'–E'**) Results of a non-parametric randomization test to show that (**D'**) ubiquitous and subcellular genes (p-value = 0) and anterior and posterior genes (p-value = 0.0018) have significantly different median 3'UTR lengths (i.e., no or little overlap of densities) and (**E'**) that

*Figure 3. continued on next page*

*Figure 3. Continued*

ubiquitous genes are significantly less conserved in their 3′UTRs than subcellular genes (p-value: 0) and posterior genes show higher conservation than anterior genes (p-value: 0.0032).

The following figure supplements are available for figure 3:

**Figure supplement 1**. Ovary gene sets have specific expression patterns during embryogenesis.

**Figure supplement 2**. Gene features of subcellular enriched mRNAs.

**Figure supplement 3**. Embryo localized mRNAs also have long, conserved 3′UTRs.

**Figure supplement 4**. Cytoplasmic but not nuclear mRNA localization requires the cytoskeleton.

**Figure supplement 5**. Posterior mRNA localization is impaired in posterior localization pathway mutants.

the localization of all new anterior and posterior candidate mRNAs is lost in colchicine-treated egg-chambers, while ubiquitously distributed mRNAs or RNA foci in the nucleoplasm, that lacks a microtubule cytoskeleton, were unaffected by the colchicine treatment (*Figure 3—figure supplement 4A–C*, *Supplementary file 8*).

However, mRNA localization requires more than an intact microtubule cytoskeleton. We therefore next investigated the localization of candidate posterior mRNAs in mutant egg-chambers that affect the localization of the known posterior mRNA, *oskar*. Posterior transport of *oskar* mRNA requires components of the exon junction complex, the RNA binding protein Staufen and an intact microtubule cytoskeleton (*van Eeden et al., 2001*; *St Johnston et al., 1989*; *Ephrussi et al., 1991*; *Hachet and Ephrussi, 2001*, *2004*; *Micklem et al., 2000*). The posterior enrichment of the selected candidate mRNAs was severely reduced in egg-chambers mutant for an exon junction complex component (*Btz¹*), that has a disrupted cytoskeleton (*Spire^RP*) or that lack Staufen (*Stau^D3*) protein. The localization of all candidate posterior mRNAs resembled the mis-localized *oskar* mRNA in these mutant conditions (*Figure 3—figure supplement 5A*).

Oskar protein is a known to be required for the assembly of functional pole plasm and the subsequent localization of mRNAs such as *nanos* (*Ephrussi et al., 1991*; *Ephrussi and Lehmann, 1992*). Therefore, we next investigated whether the novel candidate mRNAs also require Oskar protein for their posterior localization. We used genetic combinations that result in lack of Oskar protein (*osk^84/Df(3R)p^XT103*). In these Oskar protein deficient egg-chambers *oskar* mRNA is initially localized at the posterior pole at stage 9. However, the mRNA becomes successively detached from stage 10 onwards due to the lack of Oskar protein-mediated RNA anchoring (*Ephrussi et al., 1991*; *Vanzo and Ephrussi, 2002*). The novel candidates initially localized in the absence of Oskar protein at stage 9 but their posterior localization was reduced from stage 10 onwards (*Figure 3—figure supplement 5B*). Based on these experiments, we propose that the initial posterior localization of the candidate mRNAs, unlike the localization of *nanos* mRNA, is independent from Oskar protein.

Interestingly, if we completely remove posterior *oskar* RNA from the egg-chambers (*osk^A87/Df(3R) p^XT103*) (*Jenny et al., 2006*), we do not observe posterior signal for any of the novel candidate mRNAs, both at stage 9 and stage 10 (*Figure 3—figure supplement 5B*). We propose that the novel candidate mRNAs require *oskar* mRNA to initially reach the posterior pole and Oskar protein to remain stably anchored at the posterior pole beyond stage 9. The notable exception is *zpg* mRNA, that adopts posterior localization only at late stage 9/early stage 10 egg-chambers. Based on our experiments, we cannot determine whether *zpg* mRNA requires *oskar* mRNA or protein for its localization.

## mRNAs do not change transcripts but change localization during oogenesis

Our findings revealed that co-localized mRNAs share global features and have similar cytoplasmic requirements for their localization. However, as seen with *zpg*, mRNAs within gene sets differ in the precise timing and consequently regulation of their localization. We therefore investigated the

time-course of mRNA localizations in detail. In the oocyte, mRNAs can be oocyte-enriched, anterior or posterior localized (*Figure 4A*). By comparing exemplary mRNAs across oogenesis time points (*Figure 4A'*), we observed that after being oocyte-enriched mRNAs could enrich at either anterior (*Dok*) or posterior pole (*ZnT35C*), but also de-localize and show ubiquitous distribution (*exu*). Conversely, mRNAs that showed ubiquitous distribution during early oogenesis could adopt posterior localization at later stages (*aret*). These examples show that multiple combinations of mRNA distributions from early to late oogenesis are possible and mRNAs that belong to the same gene set early are not necessarily grouped together at other time points.

The same dynamics was also apparent when we compared localizations beyond the oocyte (*Figure 4B*). Maternal mRNAs that eventually enriched at the posterior pole during early embryogenesis showed any combination of mRNA distributions during oogenesis (*Figure 4B'*). For example, the anterior, ubiquitous and perinuclear mRNAs *Bsg25D*, *ssp2* and *CG14814* are all eventually enriched at the posterior pole in early embryos (*Lecuyer et al., 2007*; fly-fish.ccbr.utoronto.ca).

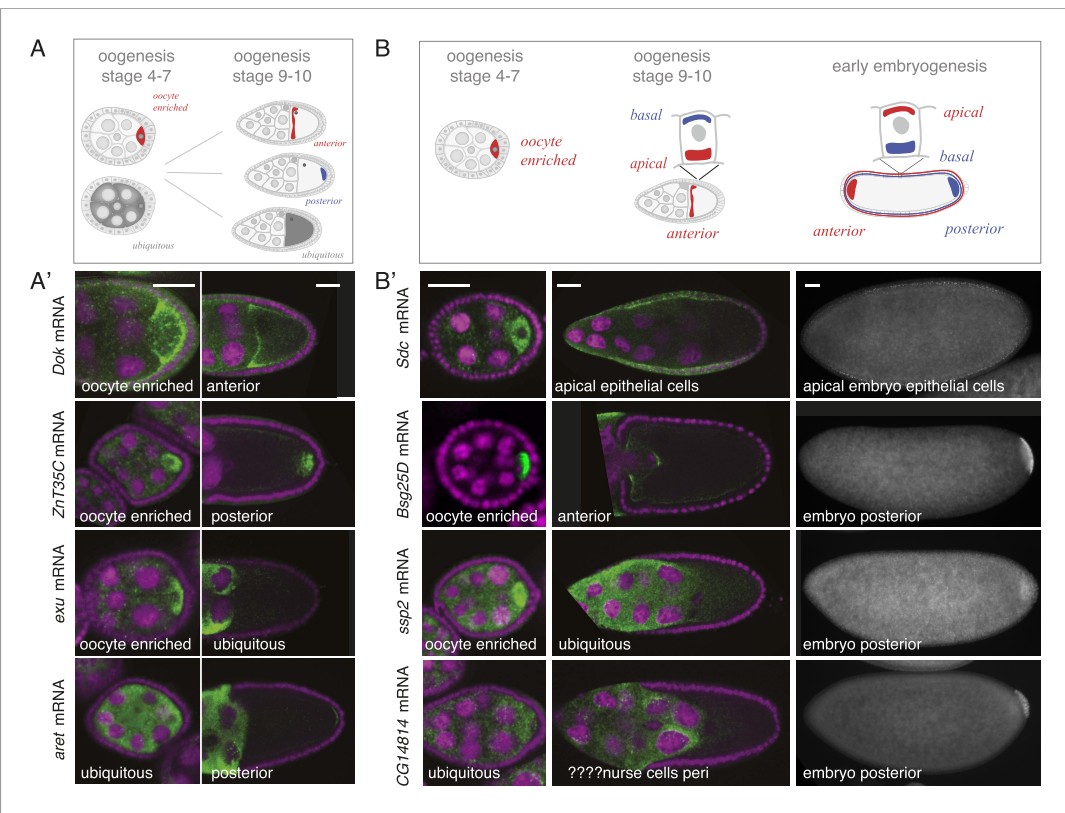

**Figure 4**. mRNA localizations change across time-points. (**A**) Schematic of changing mRNA distributions in germline cells (nurse cells, oocyte) in stage 4–7 and stage 9–10 egg-chambers. (**A'**) Exemplary mRNAs that show diverging combinations of mRNA localizations over the course of oogenesis: After initially being oocyte enriched at stage 2–7, *Dok* mRNA becomes detectable at the anterior pole, *ZnT35C* mRNA at the posterior pole and *exu* mRNA becomes ubiquitously distributed at stage 9/10. *aret* mRNA being ubiquitously distributed at stage 2–7 becomes weakly detectable at the posterior pole. (**B**) Schematic of mRNA distributions in ovary and embryonic cell types. (**B'**) mRNA expressions in ovarian and embryonic cells. All embryo data are from http://fly-fish.ccbr.utoronto.ca/. *Sdc* mRNA is localized where microtubules minus ends are enriched (*Callaini and Anselmi, 1988*; *Clark et al., 1997*; *Delanoue and Davis, 2005*) in the syncytial egg-chamber, in epithelial cells of the ovary and of the stage 4–5 embryo. *Bsg25D* mRNA is oocyte enriched, then localizes at the anterior pole in the oocyte but enriches at the posterior pole in the early embryo. Similarly, *ssp2* mRNA enriches in the oocyte during oogenesis and localizes towards the posterior pole in early embryos but during late oogenesis undergoes a ubiquitous phase. *CG14814* mRNA is initially ubiquitous, then shows perinuclear localization and in early embryos is enriched at the posterior pole. (**A'–B'**): FISH showing the RNA in green and DNA (labelled with DAPI) in magenta. Scale bar 30 μm. Embryo data are from http://fly-fish.ccbr.utoronto.ca/ (*Lecuyer et al., 2007*).

The changes in mRNA localization status within the oocyte over time prompted us to ask whether this could be explained by transcriptional regulation during oogenesis. Are we observing different transcript variants that differ in their cytoplasmic distribution? Alternative splicing was previously shown to differentially regulate mRNA localization by producing localized and non-localized isoforms of the same gene (*Whittaker et al., 1999*; *Horne-Badovinac and Bilder, 2008*). We therefore probed our stage-specific transcriptomic data for changes in gene and isoform expression. For the exemplary mRNAs shown in *Figure 4*, we could not detect significant changes in the expressed isoform (measured by RNAseq) or the 3′UTR end (measured by 3Pseq; *Figure 5A*).

We next asked whether global transcriptional changes occur that could explain differential mRNA localization during oogenesis. In agreement with results from gene expression analyses of whole ovaries measured by microarray (*Chintapalli et al., 2007*) and RNAseq (*Graveley et al., 2011*), we find that about half of the *D. melanogaster* genes were expressed at each sampled time point and the vast majority of these expressed transcripts, 85%, were detectable at every time point from early oogenesis until embryogenesis (*Figure 5—figure supplement 1B,C*). Also the expression levels across time points were highly correlated (*Figure 5B*, *Figure 5—figure supplement 1D*), suggesting that the transcriptome remained constant throughout oogenesis. Significant up- or down regulation of gene expression levels was only observed for 626 transcripts and among them are only rare examples of germline-specific transcripts (padj < 0.1, *Figure 5B*: black data points, *Supplementary files 2–4*, *Figure 5—figure supplement 1E–F*). Instead, GO-term analysis associated genes under differential expression with extracellular matrix, vitelline membrane, and cuticle formation, consistent with their expression in the somatic epithelial cells (*Figure 5—figure supplement 1E*). Across the entire oogenesis, we also could not detect shortening or lengthening of the 3′UTRs, changes in the number of transcript ends and while 55% of genes were expressed in alternative isoforms, the vast majority (>99%) of genes showed no change in isoform expression (*Figure 5C–D*, *Figure 5—figure supplement 1G–J*, *Supplementary files 5–7*). Furthermore, the ubiquitous gene set showed similar transcript diversity as subcellular genes. Therefore, changing expression levels, isoform expression, and alternative polyadenylation cannot explain the changing localization of the majority of mRNAs. The stability of the transcriptome from egg chamber formation until the onset of zygotic transcription also suggests that oogenesis is not dependent on transcriptional changes but rather on post-transcriptional regulation of the expressed transcripts, in particular through mRNA localization.

## Global changes of mRNA localization during development

A substantial portion of expressed mRNAs is localized during oogenesis. Given that the transcriptome is rather stable yet individual mRNAs show changing localizations across oogenesis, we next analysed mRNA localizations during this period globally. Within one cell, the oocyte, only few mRNAs are localized at all time points, while the majority of localizations is temporary with intermittent ubiquitous phases (*Figure 6A*). The oocyte has a highly polarized microtubule cytoskeleton that undergoes dramatic re-polarizations across oogenesis (reviewed in *Steinhauer and Kalderon, 2006*). All mRNA localizations we observed were at sites that are known to enrich for microtubule plus or minus ends. Microtubule orientation is a hallmark of cell polarity. In order to compare localizations across oogenesis stages, we categorized the localized mRNAs as being in proximity to microtubule minus- or plus- ends (*plus and minus category Figure 6B*, inset). We do not show direct association of all localised mRNAs with microtubules. However, microtubule cytoskeleton is required for RNA localization (*Steinhauer and Kalderon, 2006*) and the oocyte enriched, anterior and posterior localizations categories correspond to where the microtubule minus and plus ends are enriched (*Theurkauf et al., 1992*; *Januschke et al., 2006*).

At the different time-points of oogenesis the number of localized mRNAs varied; it was the highest at stage 2–7 and dropped to around 100 mRNAs at stage 8 and increased only slightly again towards stage 10 of oogenesis (*Figure 6B*). Also, the number of mRNAs in the minus and plus categories changed yet at different rates: during early oogenesis, the majority of mRNAs are in the minus category but the number of genes in this category rapidly dropped at stage 8 and further decreased throughout oogenesis. In contrast, the mRNAs in the plus category were increasing towards the end of oogenesis (*Figure 6B*).

To understand these trends in more detail, we plotted individual mRNAs over the course of oogenesis and clustered them by the localization category. Using this 'localization-dendrogram', we revealed that the changing localization of single mRNAs (*Figure 4*) is a global feature of localized

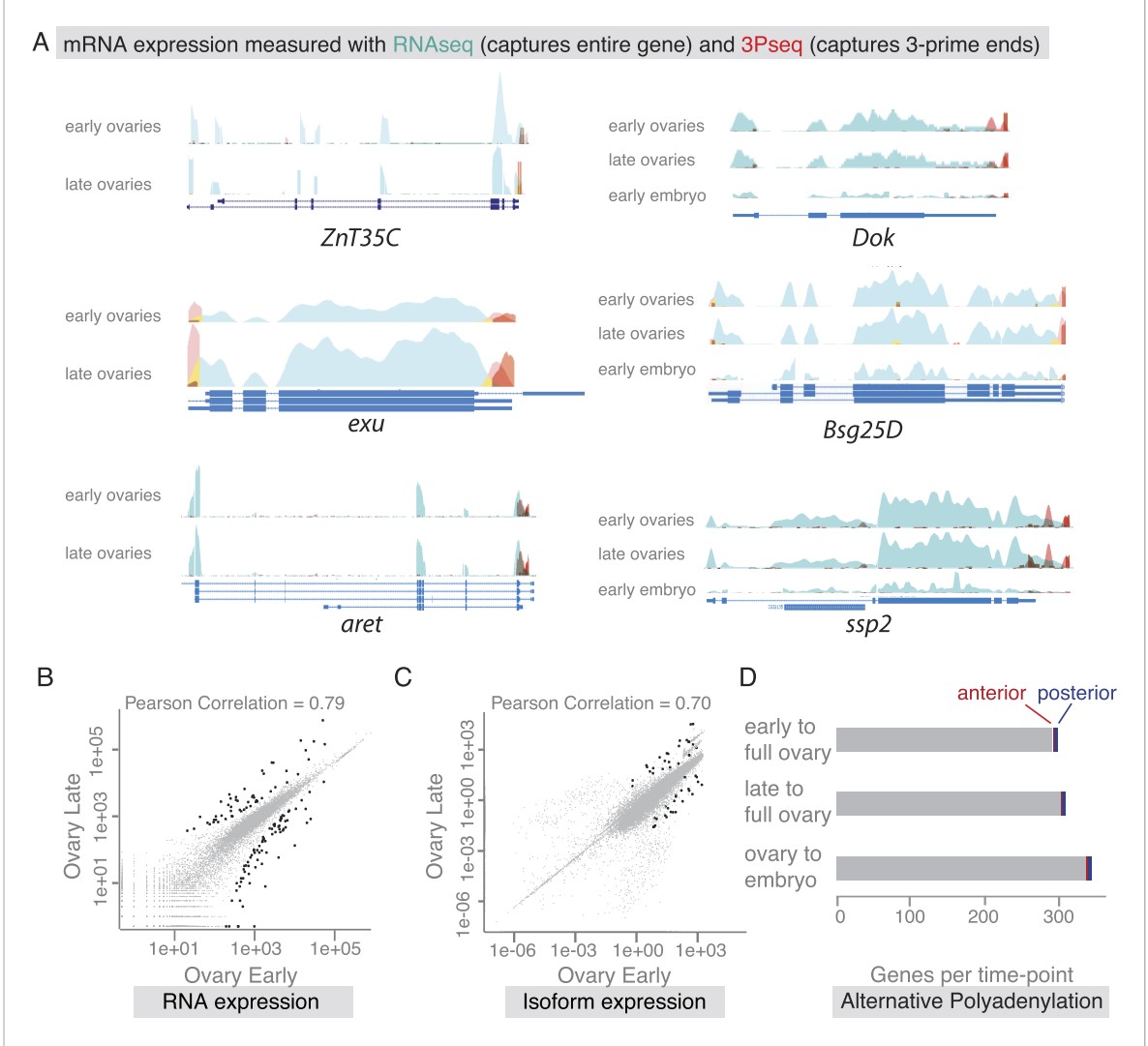

**Figure 5**. mRNA expression is stable during oogenesis. (**A**) Changing localization of *ZnT35C, exu, aret, Dok, Bsg25D* and *ssp* mRNAs across time-points (see *Figure 4*) does not coincide with a change in transcript expression: the expressed 3′UTRs (sampled by 3′prime sequencing, red) and transcript isoforms (sampled by RNAseq, green) do not change from early oogenesis to late oogenesis/early embryogenesis. (**B**) Pair-wise correlation of early/late 3Pseq data revealed that the stage-specific transcriptomes were highly similar (Pearson Correlation: 0.79); only few genes, highlighted in black, were significantly up- or down-regulated (p-value adjusted for multiple testing <0.1). (**C**) Correlation analysis of expressed transcript isoform (deduced from RNAseq data) revealed that from early to late ovaries almost no transcript-isoforms significantly changed in their expression level. Transcripts with significant changes are shown in black. (**D**) Only ~300 genes (early-full: 298; late-full: 308; full-embryo: 346) changed their mean-weighted 3′UTR length that is indicative of an alternative polyadenylation. Alternative UTR form usage across oogenesis was found for 1 (early-late oogenesis)/4 (late-full oogenesis) anterior mRNAs (red) and for 4 (early to late oogenesis)/5 (late to full oogenesis) posterior mRNAs (blue).

The following figure supplement is available for figure 5:

**Figure supplement 1**. The transcriptome shows little variation over the course of oogenesis.

mRNAs and occurred at all stages of oogenesis (*Figure 6C*). Using the localization dendrogram, we observed several groups: mRNAs that remained in the minus category at all time points, mRNAs that switched from minus category to ubiquitous distribution (this was by far the biggest category), mRNAs that switch from minus to plus category (with and without intermittent ubiquitous distribution) and ubiquitous mRNAs that become localized and affiliated with the plus category. It is noteworthy that such de novo localization of an initially ubiquitous transcript was not observed for the minus category. The dendrogram also revealed that such changes in localization occurred at all time points of

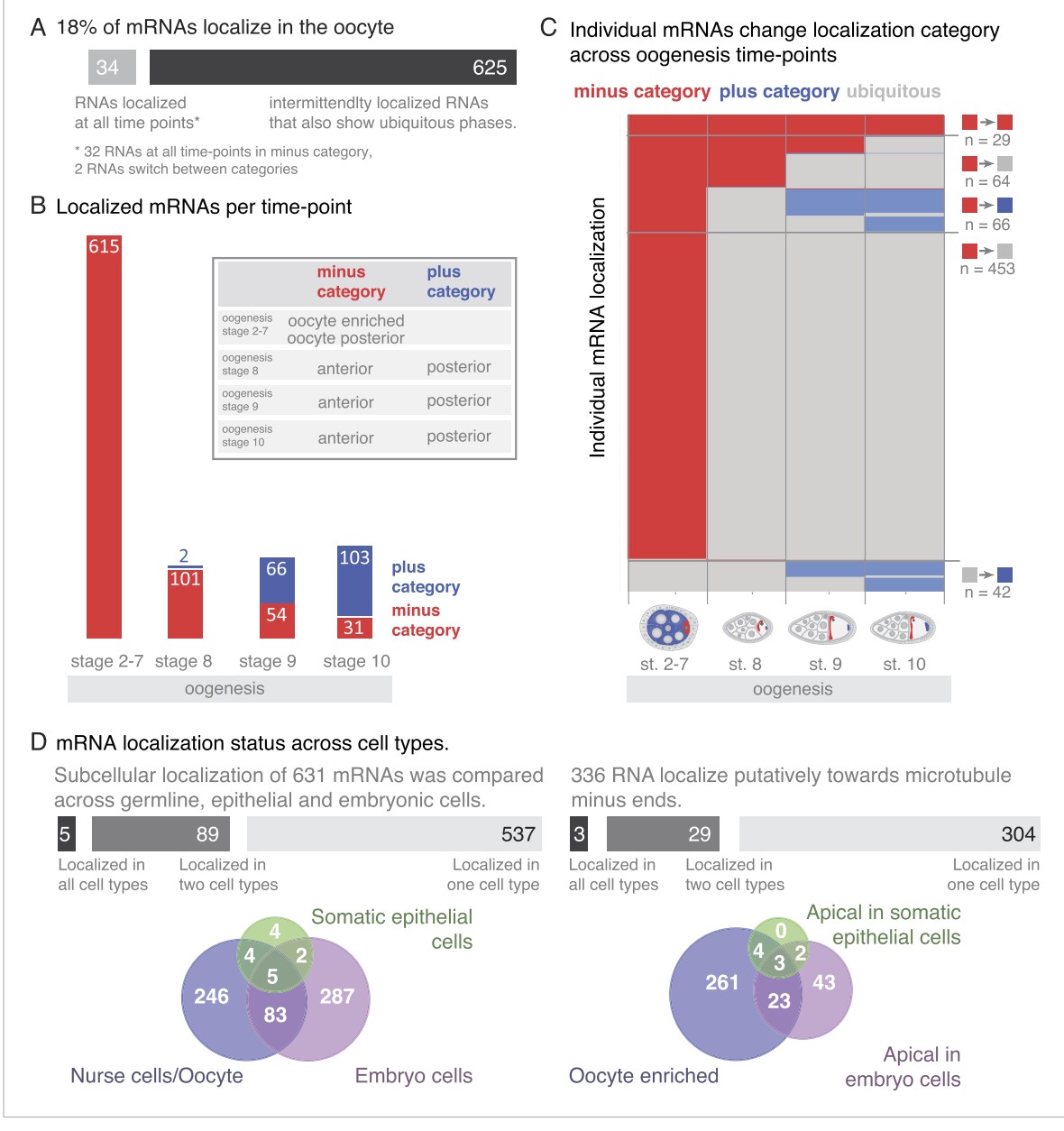

**Figure 6**. mRNA localizations are changing across cell types and within cells over time. (**A**) In the oocyte, most mRNAs of the subcellular category also have phases with ubiquitous mRNA distribution. (**B**) The number of localized mRNAs in the oocyte varies over oogenesis time points. mRNAs are grouped according to their relative localization with respect to the polarised microtubule cytoskeleton (*Steinhauer and Kalderon, 2006*). Red = mRNAs that localize where microtubule minus ends are enriched (minus category), blue = mRNAs in proximity of microtubule plus ends (plus category). (**C**) Time course of clustered single mRNA localizations. Each line represents an mRNA, indicated below are the oogenesis time-points. Localizations to the poles of the oocyte are colour-coded in red (minus category) or blue (plus category); ubiquitous phases of the mRNA are shown in grey. A summary of the trend of mRNA localizations in each cluster and the number of entries is shown to the right. (**D**) Overlap of mRNAs localized in either germline (oocyte, nurse cells), epithelial (follicle cells) and embryonic cell types shown as a Venn-diagram: Only 5 (<1%) mRNAs localized in all sampled cell types, 89 (14%) mRNAs localized in at least two cell types. The largest group in each cell type was mRNAs localized in proximity to sites known to be enriched for microtubule minus-ends (*Callaini and Anselmi, 1988*; *Clark et al., 1997*; *Delanoue and Davis, 2005*); in the early egg-chamber: oocyte-enriched; in somatic epithelial follicle cells: apical; in embryonic epithelial cells: apical. Only 3 mRNAs (<1%) showed this localization in all cell types, 29 mRNAs (9%) in two cell types.

The following figure supplement is available for figure 6:

**Figure supplement 1**. Changing localization of mRNAs in ovaries and embryos.

oogenesis: mRNAs could switch from minus category to ubiquitous distribution at stage 8, 9 or 10 or enrich in the plus category, that is, adopt posterior localization at stage 9 or 10 of oogenesis. These localization time-courses recapitulate the localization pattern of the well-characterized, singular mRNAs such as *oskar*, *gurken*, *nanos*, and *bicoid*; however, our data show that each of them occurs for multiple co-regulated mRNAs.

It has been shown that mRNA localization to the oocyte portion of the syncytial egg-chamber, to the apical side of somatic epithelial cells of the ovary and of embryonic epithelial cells in the embryo (stage 4–5) is functionally equivalent (*Bullock and Ish-Horowicz, 2001*; *Jambor et al., 2014*). Indeed, we observed mRNAs that were oocyte enriched and apical in epithelial cells (*Figure 4B' Sdc*). How general is this phenomenon? Does the majority of mRNAs localize to equivalent sites in different cell types or is it a property of singular mRNAs? To address these questions, we use again the microtubule polarity as a universal proxy of cell polarity that enables comparison of equivalent localization sites across tissues. This allows us to extend the minus and plus categories in ovaries to include data from embryos (*Lecuyer et al., 2007*). Minus category includes additionally apical sets from embryo and plus category includes pole plasm and basal embryonic enrichment categories (*Figure 6—figure supplement 1B*). We do not include pole cell annotations in the plus category since this is not a subcellular localization but rather a cell-specific expression pattern. The posterior pole plasm and anterior embryo categories reflect the polarity of embryonic body axis and do not imply any microtubule-related localization mechanism. It is for instance known that some RNAs become restricted to the posterior pole by selective degradation protection mechanism (reviewed in *Lipshitz and Smibert, 2000*). This grouping enables us to compare localization of mRNAs between life cycle stages (ovaries and embryos) and cell types (germline and epithelial cells).

To address whether oogenesis localized transcripts remain localized into early embryogenesis, we extended the localization dendrogram to stage 1–3 (maternally loaded transcripts) and stage 4–5 (after the onset of zygotic transcription) of embryogenesis (*Figure 6—figure supplement 1A,B*). This revealed that only very few of the minus and plus category mRNAs remained localized in embryogenesis: only three of the minus category mRNAs and a few more in the plus category. The plus category increased slightly at stage 1–3 of embryogenesis as a few ubiquitous oogenesis transcripts became localized. A rise of mRNAs in the minus category was only detectable at stage 4–5 of embryogenesis (apical localization) when the initiation of zygotic transcription occurs. We conclude that mRNAs are differentially localized in different developmental contexts.

Since many oocyte localized genes are also expressed in the somatic epithelium of the ovary and again during embryogenesis, we next wondered whether localization is preserved in different cell types. We took advantage of the wealth of FISH data now available for *Drosophila* and combined our data for the somatic epithelial cells and the germline (nurse cells, oocyte) cells of the ovary with the FISH screen performed on embryonic cells (*Lecuyer et al., 2007*). These screens in combination covered 9114 genes of which 1674 mRNAs showed subcellular localization at least at one time point either during oogenesis or embryogenesis and thus are 'localization competent' (*Figure 6—figure supplement 1C*). Filtering of the data sets for mRNAs that were probed by FISH in all three cell types resulted in 720 mRNAs of which only five mRNAs were localized in all three, and 89 mRNAs were localized in two cell types (*Figure 6D*). Strikingly, the data also show that with respect to microtubule polarity, only three mRNAs were in each cell type localized to the side where also microtubule minus ends are enriched (*Dok, Sdc, CG12006*; *Figure 6D*). Other types of localization, for example, nuclear RNA enrichment, had similarly minimal overlap across cell types (*Figure 6—figure supplement 1D*). The relative lack of mRNA localization to equivalent subcellular destinations indicates that while many mRNAs are localization competent, their localization appears to be cell type specific and developmentally regulated.

## Discussion

We generated a comprehensive resource, the Dresden Ovary Table (DOT, http://tomancak-srv1.mpi-cbg.de/DOT/main.html) that includes stage-specific transcriptomic and image-based RNA expression and subcellular localization data for the entire oogenesis from cystoblast division to the beginning of embryogenesis. Our resource consists of 52,000 carefully selected, annotated, stage-specific images of ovarian gene expression, and localization patterns that can be searched online or downloaded for in-depth computational analysis. The curated images are linked to 32,000 raw 3D image stacks available for interactive browsing that will facilitate further discovery. The ovary data

set is integrated with similar data on gene expression and RNA localization patterns in *Drosophila* embryos (*Tomancak et al., 2007*) enabling comparisons between tissues on a gene-by-gene basis. All visual expression patterns are described with controlled vocabularies facilitating searches and grouping of co-regulated genes. Together, this resource represents one of the most comprehensive databases of spatio-temporal gene expression patterns for two intensively studied developmental systems. The vast majority of the patterns shown in DOT are novel, often providing the very first data for computationally predicted genes. This makes the resource an excellent starting point for in-depth mechanistic studies and for enrichment analysis of gene sets generated in other genomics studies.

The global analysis of the annotation data allowed us to define gene sets of co-localized mRNAs and show that localized, particularly posterior mRNAs, have a more complex gene structure, longer and higher conserved non-coding features and higher expression levels than ubiquitous mRNAs. These properties of localised mRNAs are significant for both the oogenesis and embryogenesis data sets. Although they are not by themselves predictive of the localization status of individual mRNAs, it will be interesting to examine these properties in other cellular and developmental contexts. Our analysis, for example, predicts that the neuronal transcripts that must reach the distant synaptic compartments, would encode long, highly expressed transcripts analogous to the posterior localization gene set in the oocyte.

Similarly to embryonic cells, ovarian cells also show prevalent subcellular mRNA localizations. In contrast to the embryo system (*Lecuyer et al., 2007*), ovarian cells displayed more homogenous subcellular enrichments. With few exceptions, the candidate mRNAs localized at sites known to be enriched for either microtubule minus or plus ends. Curiously, we observed that the ovarian localized mRNAs themselves are strongly enriched for genes with cytoskeletal functions, as were localized mRNAs in the embryo (*Lecuyer et al., 2007*). This is particularly interesting in light of a recent model suggesting a self-organizing principle for the polarized cytoskeleton in mouse neurites through localized mRNAs and localized translation (*Preitner et al., 2014*). A local source of cytoskeletal proteins, for example, in the early oocyte could be beneficial to allow the rapid re-organization and growth of the cytoskeleton at the transition from early to mid-oogenesis. Next to cytoskeletal regulating factors, anterior mRNAs that localize in proximity to the meiotic oocyte nucleus were enriched for terms assigning a cell cycle regulating function. It will be interesting to investigate whether anterior mRNA localization affects meiosis, a process shown to be regulated through translational control (*Tadros et al., 2007*; *Benoit et al., 2008*; *Cui et al., 2013*; *Kronja et al., 2014*). Curiously at stage 9 and 10, we also identified mRNAs enriched in the nuclei of the oocyte. These mRNAs could either be nurse cell transcripts imported into the meiotic oocyte nucleus or else the controversial instances of transcription from the meiotic nucleus (*Saunders and Cohen, 1999*; *Cáceres and Nilson, 2005*).

Cross-tissue and time-course analyses revealed the changing mRNA localization profile during development and that the well-described, canonical examples of mRNA localization in the ovary (*Berleth et al., 1988*; *St Johnston et al., 1989*; *Ephrussi et al., 1991*; *Neuman-Silberberg and Schüpbach, 1993*) represent classes of co-regulated mRNAs. Considering that the transcriptome appears stable, we find it surprising that the same mRNA isoforms and thus the same primary sequences show such differential localizations during oogenesis. Such pervasive changes in localization status of mRNAs contradict the model that mRNAs localize through sequence encoded mRNA zipcodes (reviewed in *Medioni et al., 2012*) and that the general localization machinery is active in all cell types analysed (*Bullock and Ish-Horowicz, 2001*; *Jambor et al., 2014*). It will therefore be interesting to investigate whether specificity of mRNA localization is based on selective, cell-type-specific mRNA regulation machinery or a zipcode signal that is under specific temporal control. mRNAs can, for example, harbour two consecutively acting localization signals that direct mRNAs sequentially to opposing microtubule ends (*Ghosh et al., 2012*; *Jambor et al., 2014*). However, how one signal is de-activated and the other activated is yet unknown. Alternatively, only few mRNAs could have a zipcode for their localization and the vast majority would be co-transported with these regulated mRNAs in large transport granules. Finally, it is also conceivable that subcellular mRNAs could be locally trapped by unidentified physical properties of subcellular cytoplasmic domains or that similarly to the early embryo, some localization patterns result from protection from general cytoplasmic degradation (reviewed in *Lipshitz and Smibert, 2000*). All these mechanisms could be active consecutively or in combination during development, which could result in the observed diversity in mRNA localization.

Regardless of the specific mechanisms of mRNA transport, our genome-wide analysis shows that mRNA localization is a phenomenon contingent on the cellular context and is most likely highly regulated during development. It also highlights that oocytes do not rely on transcriptional but on post-transcriptional mechanisms to regulate gene expression, in particular (but likely not limited to) mRNA localization. Our resource enables the transition from deep mechanistic dissection of singular mRNA localization events towards systemic examination of how mRNAs transcribed in the nucleus distribute in cells and how this affects cellular architecture and cell behaviour in development.

## Materials and methods

### Mass isolation of *Drosophila* egg-chambers

Flies were grown under standard laboratory conditions, fed for 2 days with fresh yeast at 21 and 25°C. For isolation of egg-chambers, we developed a mass isolation protocol (see below) that allows us to enrich separated egg-chambers of all stages.

### RNA isolation, sequencing, and analysis

We isolated total mRNA using TRIreagent (Sigma Aldrich, Germany) from stage 1 to 7 egg-chambers, including the germline stem cells, from stage 8 to 10 egg-chambers and from total ovaries containing mainly stage 11 and older egg-chambers. Additionally, RNA from 0 to 2 hr embryos was isolated. We used two complementary mRNA sequencing approaches; standard whole mRNA sequencing (RNAseq) and a sequencing method, 3Pseq, that captures specifically the sequence adjacent to the poly(A) tail. Our 3Pseq protocol is similar to the SAPAS method described previously (*Fu et al., 2011*). In contrast to SAPAS, total mRNA was fragmented chemically, resulting in 200 nucleotide long molecules. cDNA was generated and amplified using a polyT primer terminating with a dinucleotide made of non-T followed by a random base and a 5′ template switch primer; both primers containing Illumina adaptors. This allowed us to capture each expressed polyadenylated mRNA once and thereby precisely quantify expression level (Vineeth Surendranath and Andreas Dahl, personal communication). Of the ~50 million (3Pseq) and 100 million (RNAseq) Illumina reads, we mapped 70% (3Pseq) and 90% (RNAseq) to the *D. melanogaster* release 5.52 genome with Bowtie. Quantification was done using HTSeq (*Anders and Huber, 2010*). Normalization and differential expression was done using DESeq (*Anders and Huber, 2010*). Noise thresholds of 70 and 50 counts, for RNAseq and 3Pseq respectively, were derived from observing the distributions of normalized counts. 3′UTR forms were assigned by overlaying annotated Flybase UTR forms with 3Pseq reads lying within 200 nucleotides of the annotated 3′UTR end. Alternate Polyadenylation events were called by calculating the mean-weighted UTR length (*Ulitsky et al., 2012*), a difference of 200 nucleotides in the mean-weighted lengths corresponding to two biological stages resulted in the gene being considered as undergoing Alternate Polyadenylation.

### 96-well fluorescent in situ hybridization

We used an established protocol for in situ hybridization in 96-well plates (*Tomancak et al., 2007*) with minor adaptations (see below): we added an over-night wash step after hybridization, incubate the anti-DIG antibody over night and used fluorescent tyramides for probe detection. Each experiment was evaluated and imaged using a wide-field microscope (Zeiss Axioplan Imaging, Zeiss, Germany) equipped with an optical sectioning device (DSD1, Andor Technology, UK) to generate confocal-like z-stacks.

### Annotation and database

We developed a controlled vocabulary to describe the cell types and relevant subcellular structures for oogenesis for germline and somatic cells (http://tomancak-srv1.mpi-cbg.de/cgi-bin-public/ovary_annotation_hierarchy.pl). Experiments showing no detectable FISH signal were classified as 'no signal at all stages', while experiments resulting in a homogeneous signal throughout oogenesis were classified as 'ubiquitous signal at all stages'. Gene expression patterns were imaged up to stage 10B of oogenesis after which cuticle deposition prevents probe penetration. Each pattern that did not fall in the above-mentioned classes was imaged at all stages of oogenesis in several individual egg-chambers per time point. We collected 3D images and used

custom scripts in FIJI (*Schindelin et al., 2012*) to manually select and orient representative 2D images that were uploaded to the Dresden Ovarian-expression Table (DOT) (http://tomancak-srv1.mpi-cbg.de/DOT/main). The 2D images remain linked to the original image stacks and all the raw stacks that were used to create an exemplary 2D image are available for interactive inspection using a simple image browsing cgi script. Thus, the record of each in situ experiment for a given gene consists of a set of 2D images assigned to a specific oogenesis stage and described using annotation terms selected from the controlled vocabulary. For definition of broad classifications, subclass grouping and embryo annotation class definition, see *Supplementary file 1*.

## Binary matrix

The binary matrix summarizes the data of our screen in tabular form, which facilitates access to the multidimensional image annotation data and integrates them with the RNAseq and 3Pseq data. The binary matrix is a freeze from September 2013, based on which our analyses were done. The binary matrix is provided as a flat file for independent bioinformatics investigation of the data set (http://tomancak-srv1.mpi-cbg.de/cgi-bin-public/dump_binary_matrix_ovary.pl?db=insitu_ovaries).

The matrix contains the following information for each annotated gene: the FlyBase ID; the expression levels as raw as well as normalized counts from RNAseq and 3Pseq experiments for early-, late- and full ovaries and 0–2 hr embryos; the pair-wise comparison of expression over the time course analysed, raw and normalized; the mean-weighted length indicating alternative 3′UTR expression.

The binary matrix additionally contains the annotation of FISH expression patterns. The expression terms are from the controlled vocabulary (CV). If the CV term is true its value is equal to one, otherwise it is zero. If a gene is annotated twice during the screen, the CV values are summed up and thus result in values >1.

We also provide information which clone was used to prepare the FISH probe; the classification into broad annotation classes ('no signal'; 'ubiquitous'; 'specific', see 'Results'. All reliable genes in these categories were used for the analysis and the table in *Figure 1A*); classification of specific expression patterns into subclasses ('cellular', 'subcellular', 'nuclear'); reliability status: 'reliable' and 'non-reliable' (genes probed with more than one RNA probe that resulted in conflicting annotations [n = 247], were labelled as 'not reliable'. 185 'unreliable' cases resulted from a 'no signal' vs 'ubiquitous' or 'no signal' vs 'specific' annotations, here we assume one of the probes to be non-functional. 57 'unreliable' annotations were due to different probes giving a 'ubiquitous' and 'specific' signal, respectively. One possibility is that probes were specific to different isoforms of the gene); pn-status: comparison of sequencing and FISH results (TN = true negative: genes expressed below cut-off in either RNAseq or 3Pseq and giving a 'no signal' in FISH experiments. FN = false negatives: genes expressed below cut-off in either RNAseq or 3Pseq and giving a 'ubiquitous' or 'specific' in FISH signal. TP = true positives: genes expressed above cut-off in either RNAseq or 3Pseq and giving a 'ubiquitous' or 'specific' in FISH signal. FP = false positives: genes expressed above cut-off in either RNAseq or 3Pseq and resulting in a 'no signal' FISH annotation (see *Figure 1—figure supplement 2A*)).

## GO-term analysis

For GO-term enrichment of gene sets we used the DAVID web server (*Huang da et al., 2009*). Terms or features enriched at a false discovery rate (FDR) of ≤10% and/or a Benjamini p-value of <0.1 were considered significant. Two stringencies were applied: the standard FDR cut-off (≤10%) or the more stringent 'Benjamini' p-value (≤0.1).

## Colchicine treatment and mutant analysis

Flies were fed for 15 hr at 25°C with fresh yeast paste supplemented with 50 μg/ml colchicine (*Cha et al., 2002*). The effect of colchicine on individual egg-chambers was determined by scoring the detachment of the oocyte nucleus from the anterior cortex and its migration towards the centre of the oocyte. To test posterior localization in mutants that affect *oskar* mRNA localization we used ovaries from homozygous Spire$^{RP}$ (*Manseau and Schupbach, 1989*), Stau$^{D3}$ (*St Johnston et al., 1991*), and Btz$^1$ (*van Eeden et al., 2001*) flies. Further, we analysed egg-chambers from *osk84/*Df(3R)p$^{XT103}$ flies lacking functional Oskar protein (*Lehmann and Nüsslein-Volhard, 1986*) and from *oskar*3′UTR/+; *oskA87*/Df(3R)p$^{XT103}$ flies that entirely lack endogenous *oskar* mRNA but develop past the early oogenesis arrest characteristic for *oskar* RNA null flies due to a transgenic source of oskar 3′UTR (*Jenny et al., 2006*) that is incapable posterior localization.

## Gene feature variable

For analysis of the annotated gene features, we used the flybase gff data (*D. melanogaster* release 5.52).

## 3′UTR length and conservation

For each gene, we defined the most used UTR form as the form that was most highly expressed (relative to any other forms expressed from the same gene) and which had UTR ends that overlapped by ± 200 bp with a FlyBase annotated UTR end. From this data, we extracted unique 3′UTR lengths for each gene. Sequence conservation of 3′UTRs was measured as median phyloP scores (*Pollard et al., 2010*) across all bases in the most used UTR form for 3′UTR sequence alignments across 24 *Drosophila* species (using the *D. melanogaster* UTR co-ordinates). PhyloP scores were calculated using the R package *Rphast* (*Hubisz et al., 2011*). Median UTR lengths and conservation scores were bootstrapped by re-sampling genes with replacement from selected annotation sets 100,000 times and calculating median values for each re-sample. p-values were calculated as the number of re-samples in which the annotation group with a lower median value was greater than or equal to the re-sampled median of the annotation group to which it was being compared, divided by 100,000.

## Protein interactions

A manually curated *D. melanogaster* protein–protein interaction network was downloaded from the mentha interactome database (*Calderone et al., 2013*). To test whether genes belonging to certain annotation groups participated in more protein–protein interactions within the annotation group than expected by chance, we adopted the following randomization-based approach. A random sample, the size of the number of genes in an annotation group that participate in at least one interaction in the total protein interactome, was taken from the total set of genes belonging to the protein interactome, and the number of protein interactions within this random sample was scored, minus loops. This was repeated 100,000 times to generate a distribution of the number of interactions obtained by randomly sampling the number of genes belonging to the annotation group from the total interactome. The p-value was calculated as the number of randomly sampled networks that had as many or more interactions as the real annotation group divided by 100,000.

## Protocol: mass-isolation of egg-chambers

1. Flies were fed with fresh yeast and kept for 1–2 days at 25°C.
2. Mixed sex flies were narcotized with $CO_2$ for a maximum of 5 min before proceeding to step 3.
3. Narcotized flies were immediately immersed in 4% Formaldehyde in PBS (for FISH experiments) or in PBS supplemented with 0.1% Tween-10 (for ovarian extract or total RNA isolation).
4. Flies were rapidly processed twice through a grinding mill adaptor at a fine setting (grade step '3') on a standard food processor (Kitchen Aid).
5. The ground flies were size-separated using 850, 450, and 212 µm sieves successively, resulting in a flow-through highly enriched for separated egg-chambers of all stages.
6. Collection of mass-isolated material:
   a. For FISH experiments, the co-isolation of testis and gut materials did not disturb the subsequent analysis and the material was allowed to settle by gravity and to be fixed for additional 15 min in 4% Formaldehyde, resulting in an overall fixation time of 20 min. The supernatant was then removed, the material washed twice in 1×PBS and then transferred stepwise into 100% methanol for storage at −20°C.
   b. For isolation of total RNA, we manually selected egg-chambers at early stages (germarium to stage 7, previtellogenesis), late stages (stage 9–10, postvitellogenesis), and full ovaries highly enriched for stage 11+ egg-chambers using a stereomicroscope. For each stage we collected at least 10 µl of total material that was frozen immediately.

## Protocol: 96-well plate fluorescent in situ hybridization (FISH)

1. Mass isolated egg-chambers were transferred stepwise (MeOH/PBT 3:1; MeOH/PBS 1:1; MeOH/PBS 1:3) into PBT0.1% (each wash few minutes).
2. Egg-chambers were then washed 6× in PBT0.1%, 5 min each.
3. Egg-chambers were briefly washed in PBT0.1%/Hyb 1:1.
4. Pre-hybridization of egg-chambers was done in 200 µl hybridization buffer at 55°C for 1 hr.
5. Egg-chambers were then added to a 96-well plate and hybridized over-night at 55°C in 200 µl hybridization buffer with Dextran Sulfate supplemented with 2 µl of probe.
6. 100 µl of warm Wash Buffer was added to each well and immediately removed together with probe-solution.

7. Egg-chambers were rinsed once with 150 µl of Wash Buffer and then washed four times for one hour at 55°C in Wash Buffer.
8. Egg-chambers were then washed five times for 1 hr at 55°C in 150 µl PBT0.1%, the last wash was done overnight at 55°C.
9. Egg-chambers were washed twice for 1 hr at room temperature in 150 µl PBT0.1%.
10. The primary antibody (Anti-Digoxigenin-POD Fab Fragments [Roche, Germany]) was diluted 1:200 in PBT0.1% and egg-chambers were incubated in 200 µl antibody solution overnight.
11. Egg-chambers were rinsed with 150 µl of PBT0.1% and then washed ten times for 30 min at RT in 150 µl of PBT0.1%.
12. For detection egg-chambers were incubated with Cy3-Tyramides (Perkin–Elmer, Boston Mass.) 1:70 diluted in amplification buffer for 30 min.
13. Egg-chambers were then washed ten times for 30 min at room temperature in 150 µl of PBT0.1%. DAPI, diluted 1:1000, was included in one wash step.
14. All PBT0.1% was removed and ~50 µl mounting medium was added.

## Acknowledgements

We thank Diana Selig, Jens Schmiedel, and David Seniuk for image acquisition and processing, Holger Brandl for bioinformatic services, Franziska Friedrich for drawings of egg-chambers, Anne Starkloff for webpage, Andreas Dahl, Deep Sequencing Group at CRTD/BIOTECH, Dresden for generation of the 3Pseq data, Anne Ephrussi and Daniel St Johnston for fly lines, Michael Hiller for sharing the 24 *Drosophila* species alignment. We are grateful for discussions of the manuscript to Florence Besse, Simon Bullock, Carsten Hoege, James Saenz, and Vitaly Zimyanin. VS received support from DIGS-BB. HJ and PT were supported by FP7-EU. Project: GENCODYS. PT was additionally funded by HFSP Young Investigator Grant RGY0093/2012 and by The European Research Council Community's Seventh Framework Program (FP7/2007-2013) grant agreement 260746.

## Additional information

### Funding

| Funder | Grant reference | Author |
| --- | --- | --- |
| European Commission | GENCODYS | Helena Jambor, Pavel Tomancak |
| European Research Council (ERC) | 260746 | Pavel Tomancak |
| Human Frontier Science Program (HFSP) | RGY0093/2012 | Pavel Tomancak |
| Technnische Universität Dresden | | Vineeth Surendranath |
| Max-Planck-Gesellschaft | | Helena Jambor, Vineeth Surendranath, Alex T Kalinka, Pavel Mejstrik, Stephan Saalfeld, Pavel Tomancak |

The funders had no role in study design, data collection and interpretation, or the decision to submit the work for publication.

### Author contributions

HJ, Designed and performed the experiments, Participated in the bioinformatics analysis, Wrote the manuscript; VS, Designed and performed most bioinformatic analyses including mapping, Quantification of RNAseq and 3Pseq reads, Statistical analysis of gene architecture features; ATK, Performed statistical analysis of 3'UTR length, evolutionary conservation, and protein interactome data, Co-wrote the manuscript; PM, Performed the 8000 96-well in situ hybridizations; SS, Designed an image processing script to enhance analysis of fluorescent image data; PT, Co-designed the study, Performed cross-tissue analysis, Developed database and website, Co-wrote the manuscript

### Author ORCIDs

Helena Jambor, http://orcid.org/0000-0003-3397-1842
Stephan Saalfeld, http://orcid.org/0000-0002-4106-1761
Pavel Tomancak, http://orcid.org/0000-0002-2222-9370

## Additional files

### Supplementary files

• Supplementary file 1. Gene set definitions. Definition of Gene Sets used in this analysis from the ovary and the embryo (*Lecuyer et al., 2007*) FISH annotation matrices.

• Supplementary file 2. Differentially expressed genes (early/late). Genes showing significant differential expression (padj <0.1) from early to late ovaries. Upregulated genes show log2FolgChange >0, downregulated genes show log2FolgChange ≤0.

• Supplementary file 3. Differentially expressed genes (late/full). Genes showing significant differential expression (padj <0.1) from late to full ovaries. Upregulated genes show log2FolgChange >0, downregulated genes show log2FolgChange ≤0.

• Supplementary file 4. Differentially expressed genes (full/0–2h embryos). Genes showing significant differential expression (padj <0.1) from full ovaries to 0–2 hr embryos. Upregulated genes show log2FolgChange >0, downregulated genes show log2FolgChange ≤0.

• Supplementary file 5. Differential isoform expression (early/late). Transcripts that are differentially expressed from early to late ovaries.

• Supplementary file 6. Differential isoform expression (late/full). Transcripts that are differentially expressed from late to full ovaries.

• Supplementary file 7. Changes in 3′UTR length. Shown are the mean-weighted changes in 3′UTR length across oogenesis. Values >0 indicate 3′UTR lengthening, Values <0 indicate 3′UTR shortening. Only changes affecting >200 nt are shown.

• Supplementary file 8. Effect of colchicine on oocyte mRNAs. Summary of experiments on colchicine treated egg-chambers; Shown are gene/clone name, localization in wild-type egg-chambers and mRNA appearance upon microtubule depolymerization. Data is available publicly at the DOT, the Dresden Ovary Table (http://tomancak-srv1.mpi-cbg.de/DOT/main.html).

### Major datasets

The following dataset was generated:

| Author(s) | Year | Dataset title | Dataset ID and/or URL | Database, license, and accessibility information |
|---|---|---|---|---|
| Surendranath V | 2014 | Ovary RNAseq and 3Pseq | http://www.ncbi.nlm.nih.gov/sra/?term=SRP045258 | Also publicly availalble at the DOT, the Dresden Ovary Table http://tomancak-srv1.mpi-cbg.de/DOT/main.html. |

The following previously published dataset was used:

| Author(s) | Year | Dataset title | Dataset ID and/or URL | Database, license, and accessibility information |
|---|---|---|---|---|
| Eric Lecuyer, et al., | 2007 | Data from: Global analysis of mRNA localization reveals a prominent role in organizing cellular architecture and function | http://fly-fish.ccbr.utoronto.ca/ | Publicly available at Fly-FISH (http://fly-fish.ccbr.utoronto.ca/). |

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
