## [Decision Letter]

[Editors’ note: this article was originally rejected after discussions between the reviewers, but the authors were invited to resubmit after an appeal against the decision.]

Thank you for choosing to send your work entitled “Systematic imaging reveals features and changing localization of mRNAs in *Drosophila* development” for consideration at *eLife*. Your full submission has been evaluated by K VijayRaghavan (Senior editor), Karsten Weis (Reviewing editor), and 2 peer reviewers, and the decision was reached after discussions between the reviewers. Based on our discussions and the individual reviews below, we regret to inform you that your work will not be considered further for publication in *eLife*.

The two reviewers agreed that your manuscript provides a very valuable resource for the *Drosophila* community, but felt that your work does not provide sufficient new insight into the biology of RNA localization to warrant publication in *eLife*.

Reviewer #1:

This report constitutes a valuable resource to *Drosophila* researchers studying oogenesis. The authors provide a description of 1541 FISH stained and imaged mRNAs. This and the bioinformatics analysis was a great endeavor and given their imaging expertise, the Tomancak lab is perfectly suited to tackle this project. However, it is not clear whether the authors provided any new insights into the biology of the mRNA localization. Additionally, in some places the authors oversimplify or over interpret the data (see below). Thus, the value of this manuscript beyond its value as a resource is not tangible. After correction of some of the misstated and misleading interpretations, this article could be suited as a “Resource” rather than the “Article” format. Irrespective where this will be published, the following points need to be addressed: 1) What criteria did the authors use to determine the minus and/or plus end microtubule localization? For example, in the subsection headed “Changing mRNA localization during development“, the authors state that 591 mRNAs co-localized with the microtubules minus ends during oogenesis. In the same subsection, they mention that “minus-end accumulation only re-emerged after initiation of transcription…”. How did the authors test this localization with the microtubules? The authors should clearly write whether this type of co-localization was assumed for the mRNAs tested based on previous reports or provide data showing this finding. Would it not be better and less confusing to call the localization as “anterior” or “posterior”? It would tell the same story without implying additional mechanisms of localization, which were not tested at all.

2) Figure 2—figure supplement 1: the authors conclude that the “minus-end accumulation only re-emerged after initiation of zygotic transcription of the embryo, pointing to a potential link between transcription and minus-end localization of mRNAs”. This Figure is misinterpreted as the authors neglect the fact that approximately 60% of the maternally-deposited genome is degraded in the embryo while the germ plasm enriched mRNAs are protected from degradation and remain enriched at the posterior. Thus, in this case degradation will affect the statistics in Figure 2—figure supplement 1 and the conclusion that there is a link between transcription and minus-end localization will not be valid. For example, the localization of Hsp83 mRNA at the posterior pole becomes apparent only after the activation of zygotic genome due to the degradation of un-localized Hsp83 elsewhere in the embryo. Did the authors find Hsp83 as “minus-end” localized in the stage 3–5 embryos? There are many mRNAs that localize to the posterior in a manner like Hsp83 and thus for them too this link between the minus end localization and transcription would be invalid. The authors should correct this error. Since the authors cannot account for the changes in mRNA levels due to mRNA degradation in the embryo this Figure is inherently difficult to interpret and should perhaps be removed from the manuscript.

3) In the subsection “Posterior localizations depend on oskar mRNA”, the authors identified novel mRNAs enriched at the posterior, which in the absence of osk protein are no longer localized at the posterior pole. However, instead of interpreting this finding as an indication that Oskar protein and other germ plasm components are required for the posterior localization of these RNAs, the authors interpret this finding that these novel mRNAs hitchhiked with osk mRNA to the posterior or that oskar mRNA could recruit and stabilize minus ends at the posterior thereby enabling localization of these novel mRNAs. It would have been very exciting if the authors had identified another RNA besides osk itself that could hitchhike with osk or identified a new posteriorly localized RNA that directly, and independently of Oskar protein, requires microtubules for its localization. However, apparently none of their RNAs fit this interpretation, as judged by Figure 3, where the authors show that localization of all RNAs tested is affected in *osk84/Df*, where, as the authors also show, osk RNA is properly localized. More stringent experiments would have been to determine this question in an *osk166* allele, a missense mutation (rather than the 84 nonsense allele) that affects germ plasm assembly, or other mutations downstream of Oskar that act in the germ plasm assembly pathway (Vasa, Tudor). The most definitive experiment would have been to ask whether these posteriorly localized RNAs localize to the anterior pole in oocytes and embryos expressing osk-bcd3'UTR, since in this case osk RNA localization is clearly separated from its function in germ plasm assembly.

A more likely interpretation of their result is that Oskar and the assembled germ plasm entraps the other mRNAs enriched at the posterior, as proposed previously for the posteriorly localized RNA nanos (Curr Biol. 2003 July 15;13(14):1159-68). Furthermore, there is to my knowledge no published report that suggests that Oskar has a direct an effect on microtubule polarization. Thus the fact that localization of all RNAs is affected in the oskar RNA and protein null mutants, where microtubule polarity is normal, strongly suggests that these RNAs are not directly localized by microtubule mediated transport. Failure to localize after colcemide treatment is likely a function of loss of oskar localization. It is thus misleading by the authors to interpret the defect in RNA localization after colcemide treatment as evidence for a direct role of microtubules in localization of these RNAs. The authors need to correct this misinterpretation of the data. Additional experiments such as careful analysis of the timing of RNA localization and disruption of the microtubule and actin cytoskeleton at different time points during oogenesis (see Curr Biol. 2003 July 15;13(14):1159-68) would be required to make any conclusions about the mechanisms of localization employed by the described RNAs. As these experiments are likely beyond the scope of the present analysis, the authors should avoid misleading readers.

Minor comments:

In the subsection “Widespread mRNA localization in ovaries”, the authors cite the work by Saunders and Cohen to say that the oocyte nucleus is transcriptionally active. Unfortunately, these results were shown to be incorrect by Schupbach and colleagues in Dev Biol. 2000 May 15;221(2):435-46. In this manuscript Schupbach et al. show that the construct used by Saunders and Cohen included 5'UTR sequences, which are required for localization of grk RNA to the nucleus (see also: Development. 2005 May;132(10):2345-53). To my knowledge there is not a single example that suggests that the oocyte nucleus is transcriptionally active at this stage.

Reviewer #2:

This manuscript presents a large scale in situ hybridization analysis of transcripts expressed during *Drosophila* oogenesis. The data are of high quality and provide a valuable complement to a previous study of mRNA localization patterns in the early *Drosophila* embryo. However, the analysis is almost entirely descriptive, the results are often not particularly informative, and conclusions are largely speculative.

In particular, it isn't clear why the authors find it so surprising that RNAs change localization status in different cell types and at different times. If the goal of mRNA localization is to localize proteins (or restrict their domains), it stands to reason that local requirements for the encoded proteins may be more relevant in certain cell types (e.g. oocyte vs epithelial cells) and at certain developmental times.

In addition, I'm not convinced that what the authors consider to be changes from microtubule minus-end to plus end localization or de novo plus end localization isn't simply a result of transcripts becoming anchored to the posterior cortex, such that they remain there despite changes in microtubule organization. Alternatively, what appears as de novo plus-end localization may actually be due to association of those transcripts with the nascent polar granules organized by Oskar protein, rather a specificity for microtubule plus ends. And the further increase in plus-end localization during embryogenesis does not result from a favoring of plus-end accumulation over minus-end accumulation in the embryo, but rather from the well-documented protection from degradation afforded by the germ plasm.

Finally, the authors state that the novel posterior mRNAs require the same proteins for their localization as oskar mRNA. However, their localization also requires Oskar protein so it is not possible to distinguish whether they themselves require the EJC, Staufen, etc or whether this requirement is secondary to the requirement for these proteins in localization and translation of oskar mRNA. Similarly, while the authors conclude that the posteriorly localized RNAs seem to require oskar mRNA itself, it isn't clear that the problem is the lack of oskar RNA or the lack of Oskar protein.

---

## [Author Response]

We present a genome-wide analysis of mRNA expression and localization for the entire process of oogenesis in fruit flies that in its content (imaging and transcriptomics) and depth surpasses any similar dataset in any developmental system. Our study is also the very first *comparative analysis of the “localization landscape”* between cell types and within one cell over time. mRNAs use localization elements for intracellular transport that are, analogous to protein domains, a static feature of the transcript. It has been shown that the cellular machinery, which recognizes these localization elements is universally active: expression of an oocyte-specific localization element in epithelial cells of the embryo will lead to equivalent localization (and vice versa). It is therefore highly interesting that:

1) The same mRNAs can adopt dramatically different subcellular localizations or become unlocalized in different cell types and

2) That mRNAs within one cell change their subcellular destination over time.

In comparison, a protein that harbors, for example a transmembrane domain, will remain localized to membranes in different cell types. By showing that the *transcriptome does not differ*, we rule out that changing localizations are due to changes in primary sequence of the transcript. This points to as yet un-discovered levels of regulation of mRNA distribution in cells. Importantly, such *conclusion* can *only be made based on genome wide analysis of large-scale resources* such as the ones that we present in this manuscript.

Resources are important and provide unique insights into biological problems in that they enable to generalize the properties of singular, in depth studied mechanistic phenomena. Only the data we collected enabled us to analyze genome encode features that are associated with different classes of RNA localization. We have made several striking observations, for example that RNAs that have to travel the furthest across the large germ-line cell are expressed at significantly higher expression levels. Some of the observations that we made based on the genome wide data may seemingly only confirm prior assumptions about the nature of RNA regulation (such as the length and conservation of UTRs in localized transcript). However, only our data allow us to conclude in a statistically rigorous manner that these assumptions are indeed valid. To finally be able to state using a scientific method that the signals for RNA localization (whatever they may be) are most likely encoded by the genome in the 3’UTRs?

We very strongly disagree with the argument that our resource is only useful to the narrow *Drosophila* research community. In mammalian and *Xenopus* neuronal cells almost a quarter of the expressed mRNAs enrich in neurites. Do they localize in other cell types of these vertebrates and if so how and where? Might they be expressed at exceptionally high levels? Do localized mRNAs across species share gene features similar to the ones we uncovered in *Drosophila*? These are questions of general interest provoked by our genome scale resource and its analysis. Additionally, numerous studies have attempted to use bioinformatics to predict mRNA localization signals with limited success. Our dataset breathe new life into those studies. At the same time our integrative analysis proposes that sequence independent mechanisms may play a role in localization of the major localization classes. *Thus we provide information that is essential for the RNA localization field in general.* Systematic and global approach will necessarily be descriptive and cannot provide the deep mechanistic understanding that a paper on the localization of one mRNA could grant. However, we believe that our global analysis of mRNA expression and localization in flies will benefit research in many areas beyond fruit fly oogenesis and propose new avenues of research that go beyond the established paradigms. As scientists we believe that large-scale resources are useful, enable novel insights into how genomes build cells and organisms and provide the raw material for in depth mechanistic studies. Therefore we would like to ask you to reconsider our submission as a Tools and resources article.

Reviewer #1:

*1) What criteria did the authors use to determine the minus and/or plus end microtubule localization? For example, in the subsection headed “Changing mRNA localization during development“ the authors state that 591 mRNAs co-localized with the microtubules minus ends during oogenesis. In the same subsection, they mention that “minus-end accumulation only re-emerged after initiation of transcription…”. How did the authors test this localization with the microtubules? The authors should clearly write whether this type of co-localization was assumed for the mRNAs tested based on previous reports or provide data showing this finding. Would it not be better and less confusing to call the localization as “anterior” or “posterior”? It would tell the same story without implying additional mechanisms of localization, which were not tested at all*.

Categorization of mRNA localizations:

We thank both reviewers for requesting clarification on the categorizations of mRNAs with respect to overall cell polarity. We have substantially improved the explanations for the categories given in the text, included missing references to previous work detailing the organization of microtubules during oogenesis and substantially improved the figures to increase clarity (e.g. by including schematic diagrams and simplifying the critical figures; see Figure 6 and Figure 6—figure supplement 1). For example, we have added the following excerpt to the “Global changes of mRNA location during development” subsection: “The oocyte has a highly polarized microtubule cytoskeleton that undergoes dramatic re-polarizations across oogenesis […] the oocyte enriched, anterior and posterior localizations categories correspond to where the microtubule minus and plus ends are enriched (60; 35)”.

The rationale for our categorization was to allow us to compare cell types from different developmental stages of the fly. Only through such categorization we can proceed to integrate the two large-scale datasets on RNA localization (our data and Lecuyer, 2007). The microtubule system is a very useful “universal coordinate system”. An alternative would have been the cell polarity. Both the microtubule organization (e.g. reviewed in [58]) and the cell polarity (e.g. reviewed in Suzuki and Ohno, 2006) have been well described in germline, follicle and embryonic cell types.

Why did we choose the microtubules? mRNAs in higher eukaryotes are most often transported along the microtubule cytoskeleton. In flies mRNA transport in the germline, somatic follicle cells, salivary glands and embryonic cells is equivalent. The same RNA localization signal can promote localization and the protein machinery that recognizes this “zipcode” signal is widely expressed (Bullock, 2001; Jambor, 2014).

Why did we not use the terms anterior/ posterior? Reason 1: the cytoskeleton is heavily repolarized during oogenesis. This creates the following terminological problem: mRNAs transported to microtubule minus ends at stage 2–7 would be labeled posterior, while the same transport step at stage 9–10 would be labeled anterior. Reason 2: not all cells that we analyzed localize mRNAs along the anterior–posterior axis: the somatic follicle cells and embryonic blastoderm cells (stage 4–5) localize most mRNAs along the apical-basal cortex (where “dorsal-ventral” localization would be the closest body polarity equivalent).

The reviewer also writes: “For example, in the subsection headed “Changing mRNA localization during development“*,* the authors state that 591 mRNAs co-localized with the microtubules minus ends during oogenesis.” The exact quote the reviewer is referring to is: “During *early* oogenesis, the majority of mRNAs (n = 591) co-localized with microtubule minus ends…”.

Thus, we do not claim these mRNAs are co-localized with microtubules during the entire oogenesis; in case the reviewer is questioning the use of the word “co-localization” in this sentence, we would like to point out that we did describe the relationship of RNAs and microtubules earlier in the manuscript, in the Results, as follows:

“The largest group, 591 mRNAs, was enrichment in the oocyte portion of the syncytial egg-chamber during early oogenesis (*fwe, Imp, Shroom*). At this stage the microtubule minus ends of the polarized microtubule cytoskeleton are also concentrated in the oocyte (reviewed in [58]).” Nevertheless, to be absolutely clear we have now changed the sentence to: “… during early oogenesis, the majority of mRNAs are in the minus category”.

In general we believe that the link between the organization of microtubule cytoskeleton and mRNA localization in oogenesis is so well established that it warrants the use of microtubule polarity as a proxy for comparison of RNA localization events across time and tissues.

The reviewer also writes: “In the same subsection, they mention that ‘minus-end accumulation only re-emerged after initiation of transcription…’”. How did the authors test this localization with the microtubules?” This sentence refers to the section describing our global comparisons of mRNA localization across several datasets. We state at the beginning of the section that:

*“*To address this question we use again the microtubule polarity as a universal proxy of cell polarity […]. This grouping enables us to compare localization of mRNAs between life cycle stages (ovaries and embryos) and cell types (germline and epithelial cells).”

Thus, this is an integrative analysis including published data from the Krause lab (Lecuyer, 2007) and no specific experiments testing microtubule association are involved. To be very clear we now only state that:

“This revealed that only very few of the minus and plus category mRNAs remained localized in embryogenesis: only 3 of the minus category mRNAs and a few more in the plus category. The plus category increased slightly at stage 1–3 of embryogenesis as a few ubiquitous oogenesis transcripts became localized. A rise of mRNAs in the minus category was only detectable at stage 4–5 of embryogenesis (apical localization) when the initiation of zygotic transcription occurs.”

We have removed the speculative interpretation from the manuscript altogether. We have also included a clear description of the categories (see above, see Figure 6—figure supplement 1) and made a clear reference which data is from [40].

Nevertheless, mRNA enrichment at the apical cortex in stage 4–5 embryos has been shown to be a Egl-BicD-Dynein and microtubule-dependent process (Bullock, 2001; Wilkie and Davis, 2001; Dienstbier, 2009). We therefore again argue that the co-occurrence of predominant localization sites (i.e. apical in epithelia and posterior in oocyte) with overall polarity of microtubules allows us to use this association as means to compare equivalent localization sites across tissues.

*2)*
Figure 2—figure supplement 1*: the authors conclude that the “minus-end accumulation only re-emerged after initiation of zygotic transcription of the embryo, pointing to a potential link between transcription and minus-end localization of mRNAs”. This Figure is misinterpreted as the authors neglect the fact that approximately 60% of the maternally-deposited genome is degraded in the embryo while the germ plasm enriched mRNAs are protected from degradation and remain enriched at the posterior. Thus, in this case degradation will affect the statistics in*
Figure 2—figure supplement 1
*and the conclusion that there is a link between transcription and minus-end localization will not be valid. For example, the localization of Hsp83 mRNA at the posterior pole becomes apparent only after the activation of zygotic genome due to the degradation of un-localized Hsp83 elsewhere in the embryo. Did the authors find Hsp83 as “minus-end” localized in the stage 3-5 embryos? There are many mRNAs that localize to the posterior in a manner like Hsp83 and thus for them too this link between the minus end localization and transcription would be invalid. The authors should correct this error. Since the authors cannot account for the changes in mRNA levels due to mRNA degradation in the embryo this Figure is inherently difficult to interpret and should perhaps be removed from the manuscript*.

We do not show statistics in Figure 2—figure supplement 1, we simply show a clustering of oocyte localized mRNAs and their subsequent distributions in early embryogenesis. This dendrogram only shows mRNAs that were localized in the oocyte and follows what happens to those mRNAs during stage 1–3 and stage 4–5 of embryogenesis. Consequently, many mRNAs that we either did not screen for or showed not oocyte localization will per definition not be shown in this graph, regardless of their localization mechanism. In fact, Lecuyer et al. showed many mRNAs in pole cells that we do not find to localize during oogenesis and thus apparently affects mRNAs other than the oocyte-localized mRNAs.

As described in the above section, we have now revised the figure structure, and clarified and improved the nomenclature. To avoid confusion, we have altogether removed the sentence: “Minus-end accumulation only re-emerged after initiation of zygotic transcription of the embryo, pointing to a potential link between transcription and minus-end localization of mRNAs.”

Regarding the localization of Hsp83/localization through mRNA degradation: we hope from the revised figures (Figure 6) that it becomes clear that posterior localization is included in the “plus category”, therefore Hsp83 would in our case not be in the minus category. Since the in situ in the ovaries gave no signal, the gene is entirely eliminated from the comparison of mRNAs across development. mRNA degradation of maternal transcripts occurs in two phases (Bashirullah, LIpshitz, 1999), one in the late oocytes during oocyte activation and a second phase during early embryogenesis at stage 2–3. The latter degradation pathway is mediated by SMG and does affect for example Hsp83 mRNA (Semotok, Smibert, 2005). To investigate how the many SMG-bound mRNAs (Chen, 2014) relate to oocyte localized mRNAs is very interesting. Due to the degradation/protection-from-degradation these mRNAs are enriched in the pole cells; this is an asymmetric distribution of the mRNA along the embryo, but not strictly a subcellular localization since the mRNAs is ubiquitous in pole cells and absent from the rest of the embryo. We did not include pole cell enrichment into any of our categories.

We now say: “To address this question we use again the microtubule polarity as a universal proxy of cell polarity […]. This grouping enables us to compare localization of mRNAs between life cycle stages (ovaries and embryos) and cell types (germline and epithelial cells)”.

The increase of “minus-end accumulation” that we now refer to as minus-category results primarily from apical accumulation of mRNAs in stage 4–5 of embryogenesis. These localizations arise at a time-point well beyond the SMG-mediated mRNA degradation and to our knowledge this apical mRNA localization has never been shown to result from mRNA degradation. We simply observed that an increase in the number of mRNAs in the minus category was only seen after the onset of zygotic transcription, and due to the timing (at stage 4–5), we can rule out that these are still maternal transcripts that suddenly localize again. This is a very interesting observation that could be of fundamental importance when considering how, when and where transport granules are assembled and re-modelled.

To be clear, we now say: ”The plus category increased slightly at stage 1–3 of embryogenesis as a few ubiquitous oogenesis transcripts became localized. A rise of mRNAs in the minus category was only detectable at stage 4–5 of embryogenesis (apical localization) when the initiation of zygotic transcription occurs.”

*3) In the subsection “Posterior localizations depend on oskar mRNA”, the authors identified novel mRNAs enriched at the posterior, which in the absence of osk protein are no longer localized at the posterior pole. However, instead of interpreting this finding as an indication that* Oskar *protein and other germ plasm components are required for the posterior localization of these RNAs, the authors interpret this finding that these novel mRNAs hitchhiked with osk mRNA to the posterior or that oskar mRNA could recruit and stabilize minus ends at the posterior thereby enabling localization of these novel mRNAs. It would have been very exciting if the authors had identified another RNA besides osk itself that could hitchhike with osk or identified a new posteriorly localized RNA that directly, and independently of* Oskar *protein, requires microtubules for its localization. However, apparently none of their RNAs fit this interpretation, as judged by*
Figure 3*, where the authors show that localization of all RNAs tested is affected in* osk84/Df*, where, as the authors also show, osk RNA is properly localized. More stringent experiments would have been to determine this question in an* osk166 *allele, a missense mutation (rather than the 84 nonsense allele) that affects germ plasm assembly, or other mutations downstream of Oskar that act in the germ plasm assembly pathway (Vasa, Tudor). The most definitive experiment would have been to ask whether these posteriorly localized RNAs localize to the anterior pole in oocytes and embryos expressing osk-bcd3'UTR, since in this case osk RNA localization is clearly separated from its function in germ plasm assembly*.

*A more likely interpretation of their result is that Oskar and the assembled germ plasm entraps the other mRNAs enriched at the posterior, as proposed previously for the posteriorly localized RNA nanos (Curr Biol. 2003 July 15;13(14):1159-68). Furthermore, there is to my knowledge no published report that suggests that Oskar has a direct an effect on microtubule polarization. Thus the fact that localization of all RNAs is affected in the oskar RNA and protein null mutants, where microtubule polarity is normal, strongly suggests that these RNAs are not directly localized by microtubule mediated transport*.

We thank the reviewer for pointing out the unclear description of the genetic experiments. We show that posterior localization is impaired in several mutant egg-chambers that also affect *oskar* mRNA localization and have clarified our conclusions concerning *Stau*, *Btz* and *Spire* egg chambers (see below).

We also present data analyzing mutant combination that lacks Oskar protein but the *oskar* RNA is produced. In these *osk84/Df(3R)* egg-chambers without Oskar protein, *oskar* mRNA is normally localized at stage 9 but from stage 10 onwards becomes successively detached resulting in no RNA being localized anymore in the embryo. Similarly, the candidate posterior mRNAs also localize at stage 9 but from state 10 onwards their posterior levels are reduced or entirely absent (this obviously depends on the initial amount of posterior mRNA e.g. compare the wildtype levels of *vkg* (very high) versus *PI3K21* (low at 9). We therefore conclude that the candidate posterior mRNAs initially localize in the absence of Oskar protein but, like *oskar* mRNA, their localization is reduced from stage 10 onwards. In other words, posterior localization at stage 9 does not require Oskar protein.

In contrast, in egg-chambers that produce no posterior *oskar* mRNA (*oskA87/Df(3R)*), all candidate posterior mRNAs entirely fail to localize at stage 9 and beyond. We therefore conclude that their initial localization at the posterior pole does requires *oskar* mRNA. From our initial experiments it is unclear if this is a direct or indirect effect of *oskar* mRNA. (The notable exception, which we now also discuss in more detail in the manuscript, is *zpg* mRNA. This mRNA belongs to the group of mRNAs reaching the posterior pole only at late stage 9/stage 10. In this case we can’t conclude whether *zpg* mRNA requires *oskar* mRNA or protein).

To increase the clarity of our genetic experiments, we have now:

a) Created several paragraphs: one solely on the analysis of *Stau*/*Btz*/*Spire* mutant egg chamber and a separate section on *oskar* mutants (Protein, RNA; see subsection headed “Global features of localized mRNAs”).

b) We also included stage 9 and stage 10 data for the Oskar protein mutant egg-chambers; this clearly shows the stage 9 localization of candidate mRNAs at the posterior pole (see Figure 3—figure supplement 5). We also explain the experiments and conclusions more clearly in the manuscript. We now state: “In these Oskar protein deficient egg-chambers *oskar* mRNA is initially localized […] independent from Oskar protein.”

c) Finally, we moved the figure to the Supplementary material, since we feel it is not the most critical aspect of the paper; it merely provides initial interesting observations about possible mRNA transport mechanisms. We agree with the reviewer that additional experiments would be required to clarify the mechanism of localization of these mRNAs. By de-emphasizing our initial observations we hope to shift the focus of the paper back to genomics as initially intended.

Since Oskar protein is not required for initiating candidate RNA localization at the posterior pole, proteins downstream of Oskar, such as Vasa and Tudor, are also unlikely to have a direct effect on initiating localization at stage 9.

We agree that the *osk-bcd* experiment would be interesting; it would be particularly valuable to investigate specifically the mechanisms that localize mRNAs at the posterior pole from stage 10 onwards. We would speculate that these RNAs do require factors present in the germ plasm. However, to understand if *oskar*-RNA mediated localization at stage 9 is direct or indirect the experiment would be difficult to interpret. *osk-bcd* is expressed as a transgene in the presence of wild-type *oskar* mRNA. Thus the egg-chambers will have *oskar* mRNA at both poles. Depending on which portion of the oskar mRNA would mediate the interaction with the candidate mRNAs, they would be recruited to the both poles (motif in the CDS) or just the posterior pole (motif in the UTR, as shown for *osk–osk* hitchhiking). Finally, genetic evidence suggests that *osk* mRNA is able to recruit Par1 protein and in *osk-bcd* egg-chambers indeed Par1 accumulates at both poles of the oocyte (Shulman, 2000). This strongly influences the polarity of the entire egg-chamber and would make interpretation of the results difficult. In fact, a positive feedback loop between oskar mRNA/Oskar, Par1 and microtubule plus ends was previously shown (Zimyanin, 2007). In Btz and Khc mutants that are unable to promote *oskar* mRNA localization to the posterior pole this leads to a complete loss of posterior Kinesin-beta-Gal. Conversely, ectopic localization of *oskar* mRNA ultimately results in ectopic accumulation of Par1 and also microtubule plus ends, this was shown by Zimyanin et al., 2007 (Curr Biol. 2007 February 20;17(4):353-9.).

We have now removed our interpretative statement from the Results section and only state that: “We propose that the novel candidate mRNAs require *oskar* mRNA to initially reach the posterior pole and Oskar protein to remain stably anchored at the posterior pole beyond stage 9.”

*Failure to localize after colcemide treatment is likely a function of loss of oskar localization*.

We do not think that from this experiment it is possible to conclude about direct effect of MTs on localization. And we did not and do not do so in the manuscript.

*It is thus misleading by the authors to interpret the defect in RNA localization after colcemide treatment as evidence for a direct role of microtubules in localization of these RNAs. The authors need to correct this misinterpretation of the data. Additional experiments such as careful analysis of the timing of RNA localization and disruption of the microtubule and actin cytoskeleton at different time points during oogenesis (see Curr Biol. 2003 July 15;13(14):1159-68) would be required to make any conclusions about the mechanisms of localization employed by the described RNAs. As these experiments are likely beyond the scope of the present analysis, the authors should avoid misleading readers*.

It is unclear what the reviewer is exactly commenting on. In the submitted manuscript we said:

”Transport of mRNAs towards the anterior and the posterior pole of the oocyte requires an intact microtubule cytoskeleton; accordingly, the localization of all new anterior and posterior candidate mRNAs is lost in colchicine treated egg-chambers, while ubiquitously distributed mRNAs or RNA foci in the nucleoplasm, that lacks a microtubule cytoskeleton, were unaffected by the colchicine treatment”.

We now merely state that: “…We observed that the localization of all new anterior and posterior candidate mRNAs is lost in colchicine treated egg-chambers, while ubiquitously distributed mRNAs or RNA foci in the nucleoplasm, that lacks a microtubule cytoskeleton, were unaffected by the colchicine treatment (Figure 3—figure supplement 4, Supplementary Table 8). ”

Thus, we do not state that this is a direct effect. Our conclusion from these experiments, as stated in the paper, is simply that microtubules are involved and mRNA localization is lost in colchicine egg chambers. We do not state that microtubules have a direct role.

Minor comments:

*In the subsection “Widespread mRNA localization in ovaries”, the authors cite the work by Saunders and Cohen to say that the oocyte nucleus is transcriptionally active. Unfortunately, these results were shown to be incorrect by Schupbach and colleagues in Dev Biol. 2000 May 15;221(2):435-46. In this manuscript Schupbach et al. show that the construct used by Saunders and Cohen included 5'UTR sequences, which are required for localization of grk RNA to the nucleus (see also: Development. 2005 May;132(10):2345-53). To my knowledge there is not a single example that suggests that the oocyte nucleus is transcriptionally active at this stage*.

In our conclusion, we raised two possible explanations: either the nucleus is active or the mRNAs are imported into the nucleus at stage 9. There is no evidence whatsoever for RNAs being imported into the oocyte nucleus and nuclear envelope breakdown is not supposed to occur until stage 13 of oogenesis. However, we did observe mRNAs in the oocyte nucleus. There at least have been some investigations into transcription in the oocyte nucleus and this has been subject to much debate. We agree with the reviewer that there is no good evidence so far that there is transcription from the oocyte nucleus, which is arrested in meiotic prophase and condensed into a karyosome. However, as far as we know there is also no evidence ruling out any transcription. In fact, the paper cited by the reviewer clearly states that they do not exclude the possibility that mRNAs, including *gurken*, are transcribed in the oocyte nucleus; however this is not contributing to axis formation: “while our data do not exclude the oocyte nucleus as a potential additional source of *grk* transcripts, any such contribution is not required for axis determination” (Caceres, 2005)*.*

Thus, our observation that there are many mRNAs in the oocyte nucleus (note: we did not detect *grk* mRNA in the nucleus, but interestingly *rho* which genetically interacts with *grk*) simply adds an interesting observation to this long-standing debate.

Nevertheless, for increased clarity we have now state in the Results section: “191 RNAs were detectable specifically in ovarian nuclei mostly of the endocycling, polyploid nurse cells, but also in epithelial cells and in 29 cases in the oocyte nucleus (Figure 2). The RNAs in ovarian nuclei were visible from stage 9 of oogenesis onwards and their localization changed appearance from stage 9–10 (Figure 2—figure supplement 1).” And, in the Discussion: “Curiously at stage 9 and 10 we also identified mRNAs enriched in the nuclei of the oocyte. These mRNAs could either be nurse cell transcripts imported into the meiotic oocyte nucleus or else the controversial instances of transcription from the meiotic nucleus (50; 9).”

Reviewer #2:

*This manuscript presents a large scale* in situ *hybridization analysis of transcripts expressed during Drosophila oogenesis. The data are of high quality and provide a valuable complement to a previous study of mRNA localization patterns in the early Drosophila embryo. However, the analysis is almost entirely descriptive, the results are often not particularly informative, and conclusions are largely speculative*.

In particular, it isn't clear why the authors find it so surprising that RNAs change localization status in different cell types and at different times. If the goal of mRNA localization is to localize proteins (or restrict their domains), it stands to reason that local requirements for the encoded proteins may be more relevant in certain cell types (e.g. oocyte versus epithelial cells) and at certain developmental times.

Singular mRNAs can indeed be localized and non-localized: *std* mRNA is expressed in two isoforms, only one of them encompassing a localization signal (Horne-Badovinac, 2008). *Pgc* mRNA is initially localized at the anterior pole and then changes to a posterior localization (Nakamura paper), however the mechanism is entirely unclear. Only for one mRNA, *oskar,* the mRNA regions directing opposing microtubule localization is understood in more detail (Ghosh, 2012; Jambor, 2014). To our knowledge, however, there is no study yet showing that mRNAs do change their specific subcellular localizations to the dramatic extent that we observed. In contrast, it is generally believed that mRNAs have cis-regulatory localization signals (zipcodes) that dictate their intracellular transport. Previous studies showed that specifically the cell types compared in this study express the machinery required for mRNA localization and can localize a zipcode-containing reporter to equivalent subcellular sites ([6]; Jambor, 2014). Zipcodes are sequences and integral parts of the mRNAs, similar to domains in a protein. Thus once an mRNA is expressed and the machinery is present, the zipcode should be recognized and the mRNA transported. We show that the mRNAs expressed in the oocyte show no sign of transcriptional and isoform variation during oogenesis. It follows that they should have the same motif throughout oogenesis that should promote localization. And yet, this is not the case. We were expecting to find mRNAs localized during oogenesis to show similar localization behavior in embryos. There was no published evidence that changing destinations in the cytoplasm is a global feature of mRNAs. We used the word surprising exactly once in the discussion part of our manuscript. We ask the reviewer whether disagreement on the subjective evaluation of the intuitive appeal of the data is a sound basis for dismissing a product of three years of work that generated large amounts of clearly useful data.

Previous studies describing large-scale mRNA localization were performed mainly at one time-point or in one subcellular compartment ([52]; Blower et al., ; 2007; [68]; [10]) or following mRNAs distributions during embryogenesis, which to a large degree describes changes in cellular expression patterns rather than changes in subcellular localization (40). We present the first integrative analysis of mRNA localization across cell types and developmental contexts. Surprising or not the results are what they are.

We believe that this data is important because it will strongly influence the analysis of mRNA regulation mechanisms. It will alter the way we think about zipcodes and initiate research into how they could possibly be regulated, e.g. by masking of zipcodes through proteins or by changing RNA secondary structures.

*In addition, I'm not convinced that what the authors consider to be changes from microtubule minus-end to plus end localization or* de novo *plus end localization isn't simply a result of transcripts becoming anchored to the posterior cortex, such that they remain there despite changes in microtubule organization*.

mRNAs showing de novo enrichment (either stage 9 or stage 10) show no sign of localization during early oogenesis (see Figure 6, stage 2–7, stage 8 of oogenesis). We cannot rule that indeed there is a tiny pool of mRNAs at the posterior pole that serves as a nucleation site for downstream localization. This initial amount of RNA enrichment however would be below the detection limit of light microscopy. mRNAs that change localization from oocyte enrichment (minus-end transport step, stage 2–7) to posterior accumulation at stage 9/10 are present in high concentrations in the oocyte. The fact that we cannot observe them at stage 8 of oogenesis at the posterior cortex argues against them simply remaining at the posterior cortex while the microtubule cytoskeleton is being re-organized.

*Alternatively, what appears as* de novo *plus-end localization may actually be due to association of those transcripts with the nascent polar granules organized by Oskar protein, rather a specificity for microtubule plus ends. And the further increase in plus-end localization during embryogenesis does not result from a favoring of plus-end accumulation over minus-end accumulation in the embryo, but rather from the well-documented protection from degradation afforded by the germ plasm*.

As the reviewer correctly points out, the de novo localization, that resembles for instance *nanos* mRNA localization, could indeed be posterior recruitment by the pole plasm components. We believe our shortcoming to explicitly state how we define “plus” and “minus” in the figure lead to confusion. We do not wish to claim that posterior localization at stage 10 and beyond is driven by directed microtubule movement; in fact, with the onset of cytoplasmic streaming this will most likely be impossible. We simply used “plus” to define a category of mRNA localizations.

We have now included schematics (Figure 6), simplified the figure and explicitly stated how we categorized “minus” and ‘’plus” in the Figure 6 (inset) and in the text as opposed to merely in the Supplementary Table. We state that: “In order to compare localizations across oogenesis stages we categorized the localized mRNAs as being in proximity to microtubule minus- or plus- ends (plus and minus category Figure 6, inset). We do not show direct association of all localised mRNAs with microtubules. However, microtubule cytoskeleton is required for RNA localization (58) and the oocyte enriched, anterior and posterior localizations categories correspond to where the microtubule minus and plus ends are enriched (60; 35).”